# *Acanthopharynx* Marine Nematodes (Nematoda, Chromadoria, Desmodoridae) Dwelling in Tropical Demosponges: Integrative Taxonomy with Description of a New Species †

**Alexei Tchesunov [1,*], Patricia Rodríguez García [2], Ulyana Simakova [3] and Vadim Mokievsky [3]**

1    Department of Invertebrate Zoology, Faculty of Biology, Moscow State University, 119991 Moscow, Russia
2    Centro de Investigaciones Marinas, Universidad de La Habana, La Habana 11300, Cuba
3    Coastal Ecology Laboratory, Shirshov Institute of Oceanology RAS, 117997 Moscow, Russia
*    Correspondence: avtchesunov@yandex.ru
†    urn:lsid:zoobank.org:pub:A2213892-F2B1-46FC-BF03-57439AEA59B4;
     urn:lsid:zoobank.org:act:B6F2CC58-1F62-4727-BAA3-09E00DD095D1.

**Abstract:** In the exploration of the meiofauna associated with sponges and corals in the shallows of Cuba, we investigated nine species of sponges (Demospongia), wherein 26 nematode species were revealed. Most nematode specimens (50–95% of all individuals) in all sponge samples belonged to the family Desmodoridae (order Desmodorida), followed by the family Chromadoridae (order Chromadorida). A major part of Desmodoridae is constituted by the genus *Acanthopharynx*. A statistical morphometric analysis (principal component analysis and multidimensional scaling with testing via analysis of similarities) revealed two close cohorts that differed in size and pharynx shape. Molecular genetic analyses (COI, 18S, and 28S) also distinguished two groups of specimens that corresponded to morphometric cohorts. Based on the morphometry and molecular genetics, the larger-sized group was defined as *Acanthopharynx micans* (Eberth, 1873), while the smaller-sized group was considered *A. parva* sp. n. In light of the taxonomic review of the *Acanthopharynx*, emended generic diagnosis, and the annotated list of ten valid species, *A. parva* sp. n. differed from other *Acanthopharynx* species by its peculiar shape of the pharynx (gradually widened to cardia), smaller body size, and pattern of precloacal organs.

**Keywords:** free-living marine nematodes; morphometrical analysis; molecular genetics; taxonomy

## 1. Introduction

It has long been known that marine nematodes may occur in sponges, often regularly and abundantly, e.g., see [1–4]. However, the species composition, feeding behavior, and life cycles of such nematodes are poorly known. Obligate symbiotic species are unknown, but some species (e.g., *Leptosomatum bacillatum*; see [2]) are definitely much more numerous inside than outside sponges.

As a part of our study of symbiotic micrometazoans on and in sessile macrobenthic animals in the Caribbean Basin, we examined the nematode populations of eight sponge species (Demospongia) collected on the southern coast of Cuba. The most abundant nematode taxon was found to be the genus *Acanthopharynx* (Desmodoridae). This is a rather common marine shallow-water nematode genus whose species live in sediments as well as in algal and seaweeds but have not previously been recorded in sponges. The species identification of *Acanthopharynx* is problematic, partly because the descriptions of a number of species discovered in the first part of the 20th century are incomplete or lack some minor but important details necessary for species recognition during subsequent findings. As the work of preliminary sorting proceeded, we figured that our samples of the *Acanthopharynx* may be a mixture of at least two morphospecies hardly discernible from each other via microscopical observation. Those two types differ in the shape of the

pharynx. The first type, preliminarily named *Acanthopharynx* 1, has a pharynx typical for the genus composed of anterior slender and posterior wider parts (elongated bulb). In contrast, the second type (*Acanthopharynx* 2) features a somewhat shorter pharynx that gradually widens to the posterior end similar to an elongated cone and without distinct division into narrow and wide parts. Our first assumption was that it might be an effect of the DESS fixation: due to an osmotic difference, the alimentary tract often breaks, and its parts shrink and become deformed. In addition, the two types differ in body size slightly. To confirm our assumptions, we conducted a statistical analysis of the morphometric data of *Acanthopharynx* specimens from the analyzed samples and then conducted molecular genetic testing on several individuals. Since we were faced with the necessity to compare our species with those described long ago and hence lacked some important details from the original diagnoses, we undertook a taxonomic review of the genus *Acanthopharynx*.

Therefore, this work aimed to describe and identify *Acanthopharynx* species that dwell in Cuban marine sponges together with an analysis of the morphometric variability, molecular genetic characterization of those species, and a taxonomic review of species diversity within the genus *Acanthopharynx*.

## 2. Material and Methods

### 2.1. Sampling and Morphological Observations

The sponges were sampled on 17–18 November 2019 by P.R.G. on the southern coast of Cuba in the vicinity of Cienfuegos City. The site was Ancón Beach (21°71′01.53″–21°75′31.79″ N and 79°99′39.96″–80°02′75.64″ W) at depths of 9–16 m. The sponges (Figure 1) were gathered by diving, delivered on board, broken into finer parts, and sieved. Altogether, 25 sponge samples were taken and processed. P.R.G. and José Andrés Pérez García conducted species identification of sponges. The obtained concentrates of sponge meiofauna, together with some sponge particles, were fixed with DESS. The DESS-fixed nematode specimens could be used for the preparation of glycerin slides for light microscopy; however, many specimens might have had some damage to their internal organs. For example, a pharynx may have been constricted and torn off from the intestine; this evidently resulted from an osmotic shock caused by transferring the specimens from seawater to the DESS.

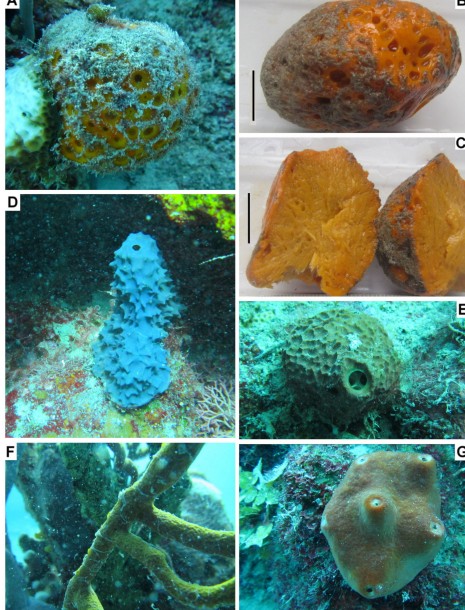

**Figure 1.** Sponge specimens harboring nematode populations: (**A**) *Cinachryrella* sp. (specimen PA13 in situ); (**B**) *Cinachryrella* sp. (specimen PA21 delivered on board); (**C**) *Cinachryrella* sp. (specimen PA21 cut); (**D**) *Aiolochroia crassa* (specimen PA2 in situ); (**E**) *Verongula rigida* (specimen PA20 in situ); (**F**) *Aplysina fulva* (PA22 in situ); (**G**) Unidentified demospongian (specimen PA8 in situ). (**B**,**C**)—scale 3 cm.

To prepare the permanent glycerine slides, the nematode specimens were placed in a mixture of distilled water, 95% alcohol, and glycerin (70:29:1, respectively) at 40 °C. After the slow evaporation of the water and alcohol, the nematode specimens were mounted in glycerin slides with a paraffin–beeswax ring, glass beads as separators, and glycine sealing along the edges of the coverslips. The observations, measuring, drawing, and picturing were conducted under a Leica DM 5000 light microscope equipped with Leica Application Suite Version 3.8.0 software and a Leica DFC 425C digital camera.

For the scanning electron microscopy, the specimens were dehydrated in a series: 20% ethanol, 40% ethanol, 60% ethanol, 80% ethanol, 95% ethanol, 95% ethanol + acetone (50:50), acetone I, and acetone II; and then they were critical point dried. Once dried, the specimens were mounted on a stub to be coated with platinum–palladium and examined with a JEOL JSM.

*2.2. Morphometric Analysis*

For the numerical morphological analysis, 96 matured specimens (53 males and 43 females) from two samples, PA13 and PA21, were measured (Supplementary Materials). Together, those two sponge specimens yielded much more *Acanthopharynx* individuals than all other sponges. The nematode males and females are treated separately. Each individual was described by a set of 34 characteristics, absolute values (direct measures), and ratios (indices). Most characteristics contain gaps in the data (lacking data for certain individuals—missing values). Altogether, we measured 53 males and 39 females. For multidimensional analysis, we exclude characters containing missing values. Eventually, we included 13 characters for males (nine absolute values and four ratios) and 21 for females. Altogether, 34 quantitative characters were measured. Two tentative morphotypes were present unequally in this set: 41 males and 30 females were provisionally ascribed to *Acanthopharynx* 1 and the rest to *Acanthopharynx* 2. Because of the inevitable missing values in the measurements, the number of characters and specimens was reduced for multidimensional analysis: 53 males were analyzed by 13 characters (9 direct measurements and 4 ratios), and 39 females were analyzed by 21 characters.

Principle Component Analysis (PCA) and Multidimensional Scaling (MDS) were used to visualize the existence of segregation of specimens by morphotypes and stations. The relative importance of characters for segregation was taken from PCA factor loadings. The significance of differences was tested by the ANOSIM (Analysis of similarities) algorithm for matrices and by parametric T-test for means of selected characters.

All of the calculations were performed using package PAST v. 4.08 (November 2021) (Øyvind Hammer, Oslo, Norway) [5].

*2.3. DNA Extraction, PCR, and Sequencing*

Altogether fifteen *Acanthopharynx* specimens were taken for analysis. Each specimen extracted from a DESS-fixed sample was identified in a microscope and then placed in a 0.6 mL tube individually and kept in 96% ethanol before processing. The DNA was isolated using 50 μL of the WLB buffer (Encyclo buffer without MgCl$_2$ (Evrogene™, Moscow, Russia), 5% Chelex solution (*v/v*), Proteinase K (5%, Qiagen™, Hilden, Germany)) during incubation at 55 °C for 60 min and 10 min at 95 °C. Sonication (30 kHg) was used before the incubation. The primer pairs and annealing temperatures used in this study are listed in Table 1. A pre-made PCR mix (ScreenMix-HS) from Evrogene™ (Moscow, Russia) (ScreenMix-HS, 0.5 μM of each primer and 1 μL of DNA) was used for the amplification. The resulting PCR product was visualized in a 2% agarose gel, purified using ethanol-ammonium acetate precipitation, and sequenced using ABI PRISM® BigDye™ Terminator v. 3.1 on Applied Biosystems (Foster City, CA, USA) DNA Analyzer 3500 ABI.

**Table 1.** Primers and annealing temperatures used in this study.

| Gene | Primer | Direction | Sequence 5′–3′ | PCR Scheme | Reference |
|---|---|---|---|---|---|
| COI | JB3F | f | TTTTTTGGGCATCCTGAGGTTTAT | at 95 °C for 15 s, annealing at 50 °C for 30 s, and extension at 72 °C for 45 s. | [6] |
|  | JB5r | r | AGCACCTAAACTTAAAACATAATGAAAATG |  | [7] |
| 28S | LSUD2A | f | ACAAGTACCGTGAGGGAAAGT | at 95 °C for 15 s, annealing at 53 °C for 30 s, and extension at 72 °C for 60 s. | [8] |
|  | LSUD3B | r | TGCGAAGGAACCAGCTACTA |  |  |
| 18S | SSUF04 | f | GCTTGTAAAGATTAAGCC | at 95 °C for 15 s, annealing at 53 °C for 30 s, and extension at 72 °C for 60 s. | [9] |
|  | 4r_nem | r | GTATCTGATCGCCKTCGAWC |  |  |
|  | MN18F | f | CGCGAATRGCTCATTACAACAGC | at 95 °C for 15 s, annealing at 50 °C for 30 s, and extension at 72 °C for 60 s. | [10] |
|  | Nem_18S-R | r | GGGCGGTATCTGATCGCC |  |  |

The chromatograms were processed using Codon Code Aligner 9.0.1 (Codon Code Corporation, Centerville, MA, USA). After primer trimming, the resulting sequences were deposited in GeneBank [11]. All sequences for the protein-coding region were checked for a stop-codon presence using TranslatorX ver. 14 (Cisco Systems, Inc., San Jose, CA, USA) [12]. Fasta-files were aligned using the MAFFT 7.308 (Biomatters, Inc., San Diego, CA, USA) [13] with the manual check and correction. Additional sequences were obtained from GeneBank.

The GeneBank assession numbers for the sequences obtained from our *Acanthopharynx* specimens are presented in Table 2.

**Table 2.** The Genebank accession numbers of sequences obtained during the current investigation.

| Specimen | Provisory Name | # Genebank Accession | | | |
|---|---|---|---|---|---|
|  |  | COI | 28S | 18S | Species |
| TAV1 | *Acanthopharynx* 1 | OP133123 | OP137144 | OP137154 | *Acanthopharynx micans* |
| TAV2 | *Acanthopharynx* 2 | OP133118 | OP137149 | - | *Acanthopharynx parva* |
| TAV4 | *Acanthopharynx* 1 | OP133125 | OP137145 | OP137156 | *Acanthopharynx micans* |
| TAV5 | *Acanthopharynx* 1 | OP133126 | OP137146 | OP137158 | *Acanthopharynx micans* |
| TAV6 | *Acanthopharynx* 1 | OP133129 | - | OP137157 | *Acanthopharynx micans* |
| TAV7 | *Acanthopharynx* 1 | OP133127 | - | - | *Acanthopharynx micans* |
| TAV14 | *Acanthopharynx* 1 | OP133131 | OP137148 | OP137159 | *Acanthopharynx* sp. |
| TAV15 | *Acanthopharynx* 2 | OP133122 | OP137150 | OP137152 | *Acanthopharynx parva* |
| TAV17 | *Acanthopharynx* 1 | OP133130 | - | - | *Acanthopharynx micans* |
| TAV19 | *Acanthopharynx* 2 | OP133119 | - | OP137153 | *Acanthopharynx parva* |
| TAV21 | *Acanthopharynx* 2 | OP133120 | - | OP137151 | *Acanthopharynx parva* |
| TAV22 | *Acanthopharynx* 2 | OP133121 | - | - | *Acanthopharynx parva* |
| TAV25 | *Acanthopharynx* 1 | OP133128 | OP137147 | OP137155 | *Acanthopharynx micans* |
| TAV26 | *Acanthopharynx* 1 | OP133124 | - | - | *Acanthopharynx micans* |
| TAV27 | *Acanthopharynx* 1 | OP133132 | - | OP137160 | *Acanthopharynx* sp. |

*2.4. Sequence Alignment and Phylogenetic Inference*

The poorly aligned sequences were eliminated from the alignments of 18S and 28S rDNA sequences using Gblocks 0.91b [14] under default and "soft" (maximum number of contiguous non-conserved positions—10, minimum length of a block—5, allowed gap positions in half of the sequences) conditions. Additionally, the BGME was used to select regions of the alignment suited for phylogenetic inference [15]. The lengths of alignments obtained here were: 372 bp (COI), 654–876 bp (18S), and 562–710 bp (28S). Since there were no significant differences between all of the three alignment analysis results for ribosomal DNA, we present only the "soft" Gblocks alignments outcomes.

We used Maximum Likelihood (to assess branch support ultrafast bootstrap [16] approximation (UFboot) and the SH-like approximate likelihood ratio test (SH-aLRT) [17]

with 10,000 bootstrap replicates were used) with the IQ-TREE multicore version 1.6.12 (IQ-TREE Team, Vienna, Austria) software [18] to reconstruct the phylogenetic relationships. The best-fit substitution model was determined by IQTREE software with the use of FreeRate heterogeneity [19,20]. The obtained phylogenetic trees were visualized with the help of FigTree 1.4.4 (Andrew Rambaut research group, Edinburgh, UK) software [21].

We applied two methods to each locus to independently delimit the evolutionary entities (tentative species) of the *Acanthopharynx*. (1) The Automatic Barcode Gap Discovery (ABGD) [22] was used with the following parameters: relative gap (X) of 1.1, minimal intraspecific distance (Pmin) of 0.001, maximal intraspecific distance (Pmax) of 0.1, K2P [23], and JC69 (Jukes–Cantor) [24] as distance metrics. (2) mPTP by selecting single-locus species delimitation with *p*-value 0.001 [25].

## 3. Results and Discussion

### 3.1. Morphometric Analysis

Aiming to reveal the possible heterogeneity of the *Acanthopharynx* collection, we undertook a morphometric analysis of a joint sample from two sponge specimens, PA13 and PA21 (see Supplementary Materials). Both sponge specimens were identified as *Cinachryrella* sp. Nematode males and females are treated separately. Based on preliminary microscopical observation, we could perceive two related and superficially similar but discernible tentative groups of specimens designated provisory as *Acanthopharynx* 1 and *Acanthopharynx* 2. The first group differs from the second by bigger average sizes (but without a distinct gap) and a pharynx consisting of two distinct regions, narrow anterior and wide posterior. The second group has a slightly smaller size and a pharynx gradually widening to the cardia without differentiation in distinct regions (Figure 2).

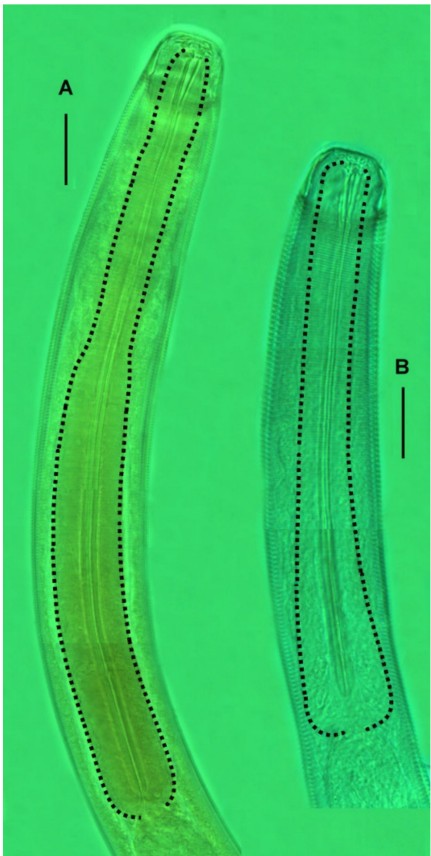

**Figure 2.** Pharynx outline (dash lined) in two varieties initially recognized: (**A**) *Acanthopharynx* 1, pharynx consists of two distinct parts, anterior narrow and posterior wide; (**B**) *Acanthopharynx* 2, pharynx elongate conoid. Scale bars 20 μm.

The results of the parametric and non-parametric analyses are congruent: *Acanthopharynx* 1 and *Acanthopharynx* 2 are separated based on a set of characters irrespectively to the sponge sample. The multidimensional scaling method, using Euclidean distances as a measure of the similarity among the specimens, shows a clear division of both males and females into two groups corresponding precisely to *Acanthopharynx* 1 and *Acanthopharynx* 2 (Figure 3A,B). The only exception is one female that fits into the group *Acanthopharynx* 1 despite being initially assigned to *Acanthopharynx* 2. At the same time, no overall separation of the nematodes between the samples PA-13 and PA-21 was revealed. ANOSIM confirms significant differences between the morphotypes (0.0001) and no differences between the samples (Table 3). The contribution of the variable "sample" is statistically insignificant.

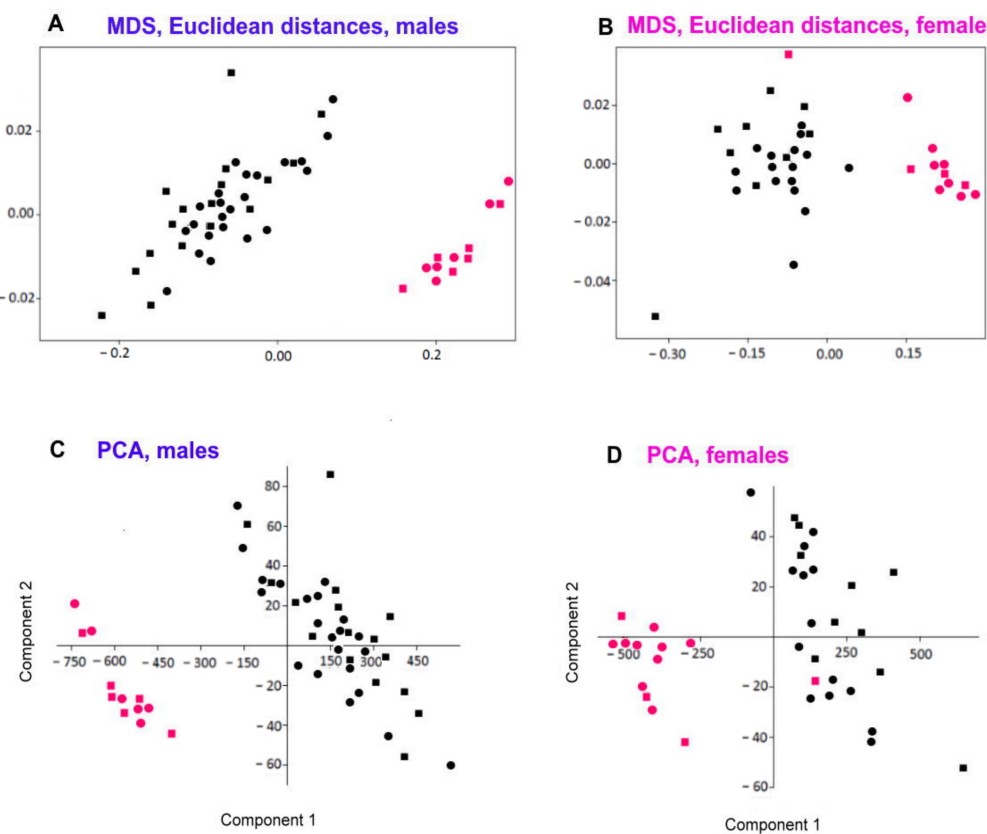

**Figure 3.** Graphic representation of morphometric heterogeneity of the *Acanthopharynx* individuals from the joint sample of PA13 and PA21 sponge specimens. (**A**,**B**)—multidimensional analysis, discrimination of two assemblages of males (**A**) and females (**B**) from samples PA13 (circles) and PA21 (squares) belonging to groups *Acanthopharynx* 1 (**black**) and *Acanthopharynx* 2 (red). (**C**, **D**)—principal component analysis of two assemblages of males (**C**) and females (**D**) from samples PA13 (circles) and PA21 (squares) belonging to groups *Acanthopharynx* 1 (**black**) and *Acanthopharynx* 2 (red).

**Table 3.** The results of Two-way Analysis of Similarities (ANOSIM) for males ($n = 53$) and females ($n = 39$). Samples and groups were the same as in Figure 2. The test statistics R (tend to 1 as differences increase) and significance $p$ were calculated using PAST v.4.10.

| Males | | |
|---|---|---|
| Factor Sample | R: 0.011257 | $p$ (same): 0.3374 |
| Factor Group | R: 0.98064 | $p$ (same): 0.0001 |
| **Females** | | |
| Factor Sample | R: 0.17168 | $p$ (same): 0.0466 |
| Factor Group | R: 0.82416 | $p$ (same): 0.0001 |

The Principle Component Analysis (PCA) produces the same result for the same sample and a set of characters: the discretion of the first component according to the tentative morphotypes and the spread in the values of the second component (Figure 3C,D). The first component describes 99.1% of the total variability for males, and the second component is just 0.8%. The most important character of the first component is the body length (loading 0.98), and that in the second component, it is the pharynx length (loading 0.97). The first component describes 98.6% of total dispersion in females. Females are discriminated by the body length, head–vulva distance, and pharynx length together with the "b" index.

The analysis of the solitary characters of the males demonstrates a clear separation between the two groups indicated by the body (Figure 4A,B) and pharynx (Figure 4C,D) lengths and the spicule length along the arch (Figure 4E) and the chord (Figure 4G). Females are distinctly separated by the index "b" (Figure 4F) and pharynx length (Figure 4D), body length (Figure 4B), and head–vulva distance (Figure 4H).

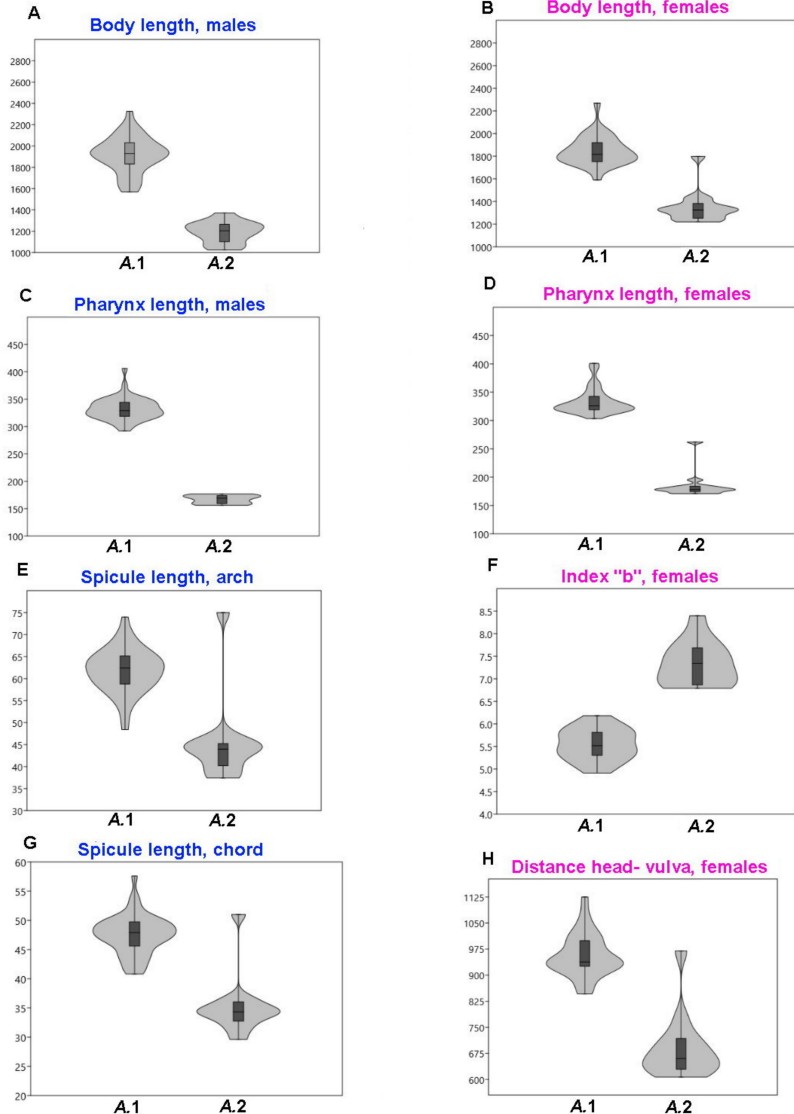

**Figure 4.** The examples of the solitary characters analyses of the two assemblages of males (**left**) and females (**right**) from samples PA13 and PA21 belonging to groups *Acanthopharynx* 1 (*A*.1) and *Acanthopharynx* 2 (*A*.2): (**A**) Body length, males; (**B**) Body length, females; (**C**) Pharynx length, males; (**D**) Pharynx length, females; (**E**) Spicule length, arch; (**F**) Index "b", females; (**G**) Spicule length, chord; (**H**) Distance head—vulva, females.

In conclusion, the mature individuals of both genders are clearly separated into two morphotypes. The best discriminant character is the pharynx length for both sexes, with 280 μm as the critical value (the gap for males is even bigger, from 180 to 280 μm). The second character is the body length for males (critical value 1400 μm), then index "b" for females (critical value 6.5). Each of these measurements discriminates specimens well, even taken alone. The differences are summarized further in "Differential diagnosis of *Acanthopharynx parva* sp. n." (see Section 3.3).

### 3.2. Molecular Genetics

The amplification with at least one primer pair was successful for twelve (out of 20) individuals from four different samples (PA13, PA21, PA22, PA24). After the lower-quality sequences were discarded, the twelve sequences for COI, seven for 28S, and ten for 18S were obtained (Table 2). No internal stop codons were found in COI sequences.

After primer and poorly read-end clipping, the length of the COI sequences was 372 bp. The number of differences in COI within the *Acanthopharynx* 1 was 0–52 bp (100–86% of identity). For *Acanthopharynx* 2, it spans 0–2 bp (100–99% of identity). Thus, the *Acanthopharynx* 1 comprises two groups of sequences. The number of substitutions within each sequence was equal to or less than 2 bp, with less similarity between the groups (86%, 52 substitutions).

The COI alignment (including Genbank data) comprised 24 sequences and 372 sites, of which 181 were variable (160 parsimony informative). The sequences of *Acanthopharynx* 1 and *Acanthopharynx* 2 formed three clades sister to *Acanthopharynx micans* and *Acanthopharynx* aff. *micans* representatives from Genbank with high to moderate UFboot/ SH-aLRT values (Figure 5). The clade of *Acanthopharynx* 1 formed two distinct clades, which is well matched with the similarity data (86% similarity between mentioned clades). The ABGD analysis with both distance metrics showed that an evident barcoding gap exist and all three mentioned clades of the *Acanthopharynx* are genetically distinct. The mPTP method based on our phylogenetic tree recovered all three clades of the *Acanthopharynx* as independent evolutionary entities (*p* = 0.001).

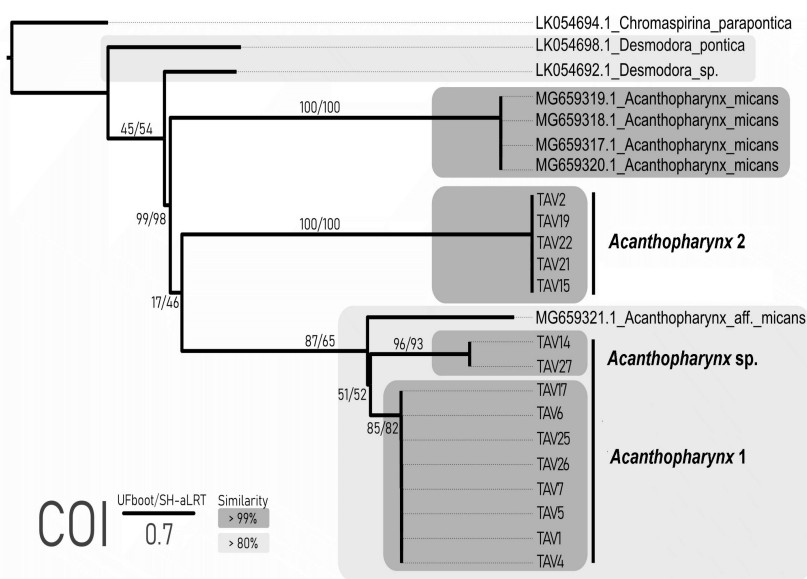

**Figure 5.** COI ML Phylogenetic tree. Branches are shown with the UFboot/SH-aLRT support. The sequences with similarities ≥80% and 90% are indicated with light and dark grey.

The lengths of the 18S sequences obtained after primer clipping varied within 822–894 bp for *Acanthopharynx* 1 and 847–863 bp for *Acanthopharynx* 2. All sequences of *Acanthopharynx* 1 were highly similar (99.6–100%). The similarity within the second species comprised 98.7–98.9%. The similarity between both morphospecies was less than 96%. The alignment

of the 18S locus used for phylogeny reconstruction included 31 sequences with a length of 799 bp, of which 246 were variable and 138 were parsimony informative. All *Acanthopharynx* species are grouped into one highly supported clade. The phylogenetic relationships of the other genera are not clear. Both morphospecies formed distinct clades sister to other *Acanthopharynx* species (Figure 6A). The results of ABGD and mPTP analysis also support the distinctiveness of these two clades.

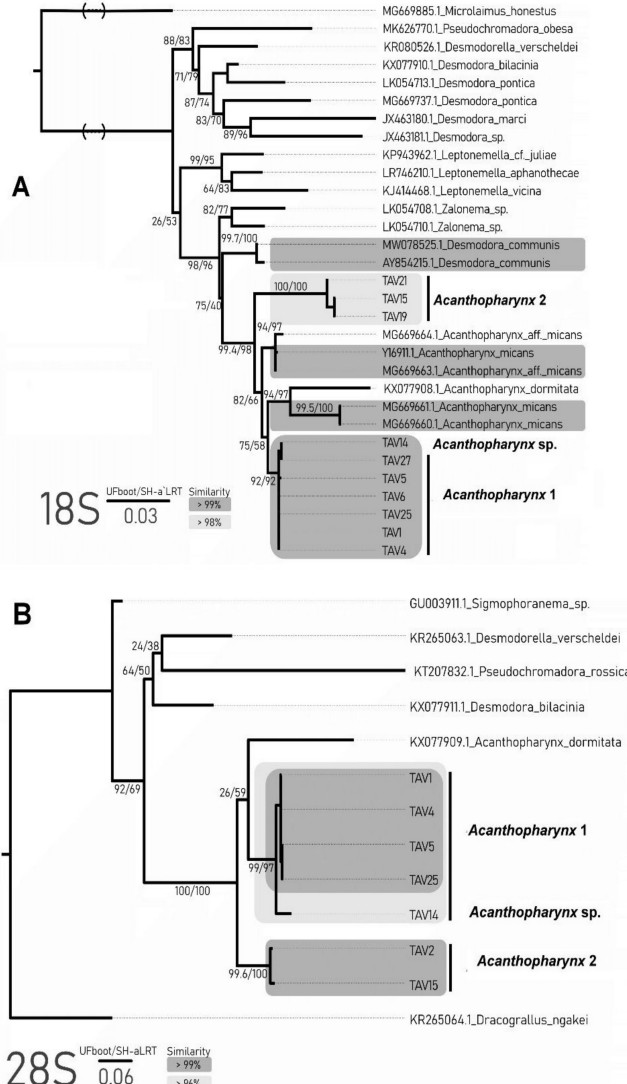

**Figure 6.** Phylogenetic trees based on nuclear sequences: (**A**)—18S ML Phylogenetic tree. Branches are shown with the UFboot/SH-aLRT support. The sequences with similarities ≥98% and 90% are indicated with light and dark grey. (**B**)—28S ML Phylogenetic tree. Branches are shown with the UFboot/SH-aLRT support. The sequences with similarities ≥96% and 90% are indicated with light and dark grey.

The obtained sequences for 28S were shorter than 18S and comprised 666 to 692 bp for *Acanthopharynx* 1. The length of *Acanthopharynx* 2 sequences was 669 bp and 667 bp, with 98.1% similarity between them. The similarity within the first species varied from 96.1% to 99.9%, which is explained by the higher dissimilarity of one sample (TAV14). Without this individual, the similarities between other specimens were more than 99%. The 28S alignment was 655 bp long, with 332 variable sites, of which 200 were parsimony informative. The phylogenetic reconstruction showed that *Acanthopharynx* species formed one clade sister to other genera (Figure 6B). Two well-supported clades corresponded to morphospecies *Acanthopharynx* 1 and 2. The third one (TAV14) belongs to the *Acan-*

*thopharynx* 1 clade but is distinguished from its other representatives. The ABGD and mPTP analyses support the presence of three clades of *Acanthopharynx* apart from the Genbank data.

According to the species delimitation and phylogenetic tree analysis based on three loci (18S, 28S, COI), the studied specimens formed at least two distinct clades: *Acanthopharynx* 1 and 2. Two loci (28S, COI) support the existence of the third clade within the *Acanthopharynx* 1: specimen TAV14 and TAV27 as a distinct evolutionary entity. These two are probably cryptic species.

According to the molecular analyses, the *Acanthopharynx* 1 species fits into the same clade with specimens designated as *Acanthopharynx* aff. *micans* (COI) or with specimens designated as *Acanthopharynx micans*, *Acanthopharynx* aff. *micans* and *Acanthopharynx dormitata* (18S). Since *Acanthopharynx* 1 differs distinctly from *A. dormitata* in structural characters and morphometrics and at the same time does not differ or scarcely differ from descriptions of *A. micans* (see below in more detail), we recognize *Acanthopharynx* 1 as *Acanthopharynx micans*.

*Acanthopharynx* 2 is clearly separated from *Acanthopharynx* 1 in all three loci (COI, 18S, 28S). Since *Acanthopharynx* 2 also differs from *Acanthopharynx* 1 and other *Acanthopharynx* species by the shape of the pharynx and some morphometric characters, we consider it a new species for science and designate it as *Acanthopharynx parvus* sp. n. (see taxonomic part below).

As for individuals TAV14 and TAV27, prior to DNA extraction, they were examined under a light microscope and designated as *Acantopharynx* 1. Since no particular features in their morphology and morphometry were revealed, the status of these two specimens remains unresolved.

### 3.3. Taxonomy

*Acanthopharynx* is one of the earliest established genera of free-living marine nematodes [26]. Since then, eighteen nominal species have been discovered worldwide [27,28]— yet some of those species were later considered junior synonyms or referred to other genera. Genus *Acanthopharynx* can be rather easily recognized from other genera of Desmodoridae owing to numerous apical (cephalic and subcephalic) setae on the head and peculiar pharynx shape. Instead, identifying species may be difficult because the species differ from each other in fine details, such as supplementary papillae and pores, which are often missing in earlier descriptions. Some species described before the latter half of the twentieth century are treated as *species inquirendae*, which means their validity may be restored after a thorough redescription based on specimens from the type locality.

The descriptions of species issued later, starting from *Acanthopharynx denticulata* Wieser, 1954, are characterized by a high level of detail and allow us to understand the morphology of the *Acanthopharynx* [8,29–31]. Thus, Leduc & Zhao [8] constructed a schematic pattern of sensilla on the head capsule based on SEM observation; they also provided molecular sequences (SSU and D2-D3 of LSU) of *A. dormitata* and showed that *Acanthopharynx* forms a basal clade to the Desmodorinae/Spiriniinae. Though Wieser [29] provided a dichotomous key for the identification of eight species known at that time, the lack of necessary details and hence the nonuniformity of morphological data between formerly and later described species encumber the development of an upgraded species identification key.

Order Desmodorida de Coninck, 1965

Family Desmodoridae Filipjev, 1922

Subfamily Desmodorinae Micoletzky, 1922

Genus *Acanthopharynx* Marion, 1870

(=*Xanthodora* Cobb, 1920; *Brachydesmodora* Allgén, 1932)

Diagnosis

amended after Tchesunov 2014 and Leduc & Zhao 2016 [8,32].

Desmodoridae, Desmodorinae. Cuticle distinctly annulated between the cephalic capsule and terminal tail cone; no lateral differentiation. Cephalic capsule non-articulated,

smooth. Labial region is bordered by the cephalic capsule by a fine suture. The anterior sensilla is concentrated close to the apex and is composed of six inner labial papillae (may be indistinct), six outer labial papillae, and four cephalic setae; a number (about ten or twelve) of subcephalic setae installed in the same crown with cephalic setae; the subcephalic setae are nearly equal in length and shape to cephalic setae and hardly distinguishable from them. Minute somatic setae arranged in irregular rows along the body. Amphideal fovea spirally coiled in one turn, round or rarely longitudinally oval in outlook; the cuticle spot in the center of the fovea differs from the cuticle outside the fovea. Buccal cavity armed with a solid movable dorsal tooth and two transversal ventrolateral rows of minute denticles; additional denticles may be present. The pharynx is muscular and has a thick internal cuticular lining along its entire length; the posterior 40–60% part is either sharply swollen as an elongated bulb or gradually swollen to its posterior end. Precloacal midventral supplementary organs usually present as a series of tiny papillae or pores and larger papilla close to the cloaca. Spicules arcuate, slightly knobbed. Gubernaculum as a short bar contiguous to spiculum. Tail short, conical, slightly bent ventrally, with a smooth terminal cone.

Type species *Acanthopharynx affinis* Marion, 1870.

Annotated species list (valid species names **in bold**)

(1) ***Acanthopharynx affinis* Marion, 1870.** Marion, 1870a: 36–37, Plate K, Figure 4–4b [26]; Mediterranean. Schuurmans Stekhoven 1942:243–244, Figure 13A–C [33]; Mediterranean.

(2) *Acanthopharynx brachycapitata* (Allgén, 1947). Allgén, 1947: 148–149, Figure 51a–c [34] (only female) (as *Desmodora brachycapitata*); Gulf of Panama. Gerlach 1963: 77 [35] (transfer to *Acanthopharynx*). Verschelde et al. 1998:82 [36] (**species inquirenda**).

(3) ***Acanthopharynx denticulata* Wieser, 1954.** Wieser, 1954a: 36, Figure 113a–d [29] (males, females, juveniles); North Chile, littoral algae, sheltered. Armenteros et al. 2014: 5–8, Figures 1A–D and 2, Table 1 [31] (males, females, juveniles); Cuba, south coast, 2 m, sand flat (lapsus *denticulatus*). Cuban males differ from Chilean males in some measurements such as body length (1234–2780 versus 2170–2780 μm), index b (5–7 versus 8.1–9.7), index c (16–21 versus 22–31)—however, Armenteros et al. [31] consider the differences as intraspecific.

(4) ***Acanthopharynx distechei* Decraemer & Coomans, 1978.** Decraemer & Coomans, 1978:515–519, Figure 2A–E [30] (male, female, juveniles); Great Barrier Reef, Lizard Isl., mangrove swamp.

(5) ***Acanthopharynx dormitator* Leduc & Zhao, 2016.** Leduc & Zhao, 2016: 908–916, Figures 1–6 [8] (males, females); New Zealand, Wellington, lower intertidal zone, red seaweed partially covered in sediments on boulders.

(6) ***Acanthopharynx japonica* Steiner & Hoeppli, 1926.** Steiner & Hoeppli, 1926: 551–555, 568–569, Figures A–F [37]; Pacific coast of Japan.

(7) *Acanthopharynx merostomacha* (Steiner, 1921). Steiner, 1921: 52–54, Tafel 3, Figure 12a–c [38] (one juvenile) (as *Desmodora merostomacha*); Red Sea. Schuurmans Stekhoven 1943: 363 [39] (transfer to *Acanthopharynx*). Allgén 1951: 300–301, 389 [40] (male, female); Pacific (Honolulu, California, Bay of Panama). Allgén, 1959: 114 [41]; Falkland Islands, South Georgia. Verschelde et al., 1998: 82 [36] (**species inquirenda**).

(8) ***Acanthopharynx micans* (Eberth, 1863) Marion, 1870.** Eberth, 1863: 4, 6, 12, 28, Table 1, Figures 1–5 [42] (male, female) (*Odontobius micans*); Mediterranean, Nizza. Marion 1870: 6 [26] (transfer to *Acanthopharynx*). Micoletzky, 1924: 148–151, Figure 1a–b [43]; Mediterranean, Red Sea. Schuurmans Stekhoven 1942: 244 [33] (*Acanthopharynx marioni*). Wieser 1954: 199, Abbildung 13 [44] (males); Mediterranean. Gerlach 1963: 93–94, Table 9, Figure g–k [35] (male); Maldives. Schuurmans Stekhoven 1950: 121, Figure 71A–C [45] (male) (as *Acanthopharynx seticauda*); Mediterranean.

(9) ***Acanthopjharynx micramphis* Schuurmans Stekhoven, 1942.** Schuurmans Stekhoven, 1942: 245–247, Figure 14A–C [33] (male, juveniles); Mediterranean, Ibiza. Despite presence of a male in the type series, all the body dimensions are given for a juvenile.

Nonetheless, the males of the species can be recognized owing to the small amphideal fovea (14% cbd) and short conical tail (c' 1.75).

(10) *Acanthopharynx nuda* **(Cobb, 1920) Gerlach, 1963.** Cobb, 1920: 317–318, Figure 98 [46] (male, female) (as *Xanthodora nuda*); Indonesia, Larat Island. Gerlach 1963: 94 [35] accepted *Xanthodora* as a synonym of *Acanthopharynx*.

(11) *Acanthopharynx parva* sp. n. Present paper.

(12) *Acanthopharynx perarmata* **Marion, 1870.** Marion, 1870: 34–35, Plate K, Figure 1–1f [26] (female); Mediterranean. Schuurmans Stekhoven 1942: 245 [33]; Mediterranean, Naples. Schuurmans Stekhoven 1950: 118–120, Figure 69A–C [45] (male); Mediterranean, Villefranche. Despite the incompleteness of the original and subsequent descriptions, the species can be recognized by a very short tail and three prominent wartlike papillae (orig. "excrescences") on the ventral side of the tail.

(13) *Acanthopharynx rigida* **Schuurmans Stekhoven, 1950.** Schuurmans Stekhoven, 1950: 120–121, Figure 70A–C ([45] male); Mediterranean, Villefranche.

(14) *Acanthopharynx setosissima* Schuurmans Stekhoven, 1943. Schuurmans Stekhoven, 1943: 362–363, Figure 31A–B [39] (female); Mediterranean, Alexandria. The original description based on a single female is incomplete and does not allow for the exact identification of this species. Hence, we consider this species as a **taxon inquirendum.**

(15) *Acanthopharynx similis* (Allgén, 1932) Gerlach, 1963. Allgén, 1932: 463–464, Figure 23a–b [47] (only juvenile) (*Desmodora* (*Brachydesmodora*) *similis*); Campbell Island, slime of cyanophycean algae. (=Allgén 1932: 133–135, Figure 21a–c [48]); Gerlach 1963: 77 [35] (transfer to *Acanthopharynx*). Verschelde et al. 1998: 82 [36] (**species inquirenda**).

*Acanthopharynx micans* (Eberth, 1863) Marion, 1870.
Figures 7–10, Tables 4 and 5.

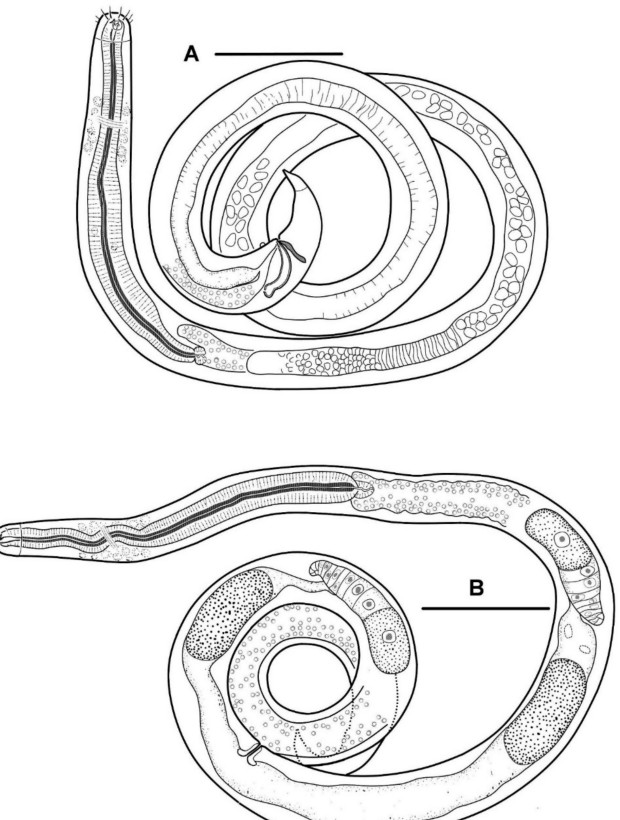

**Figure 7.** *Acanthopharynx micans*, entire (sample PA13, sponge *Cinachrya* sp.): (**A**)—male. (**B**)—female. Scale bars 100 μm.

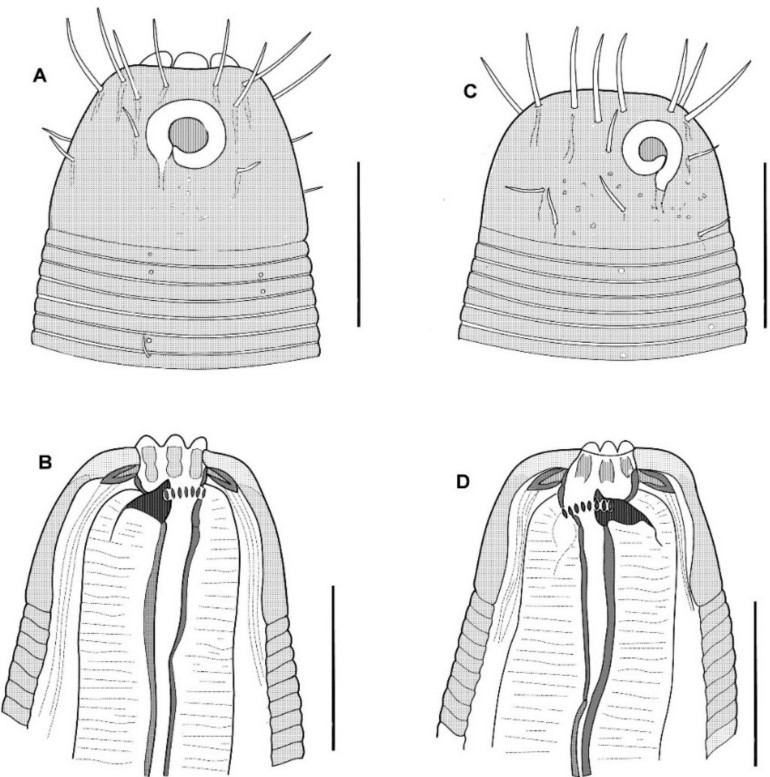

**Figure 8.** *Acanthopharynx micans,* anterior ends (sample PA13, sponge *Cinachry*a sp.): (**A**)—male, surface view. (**B**)—male, optical section. (**C**)—female, surface view. (**D**)—female, optical section. Scale bars 20 μm.

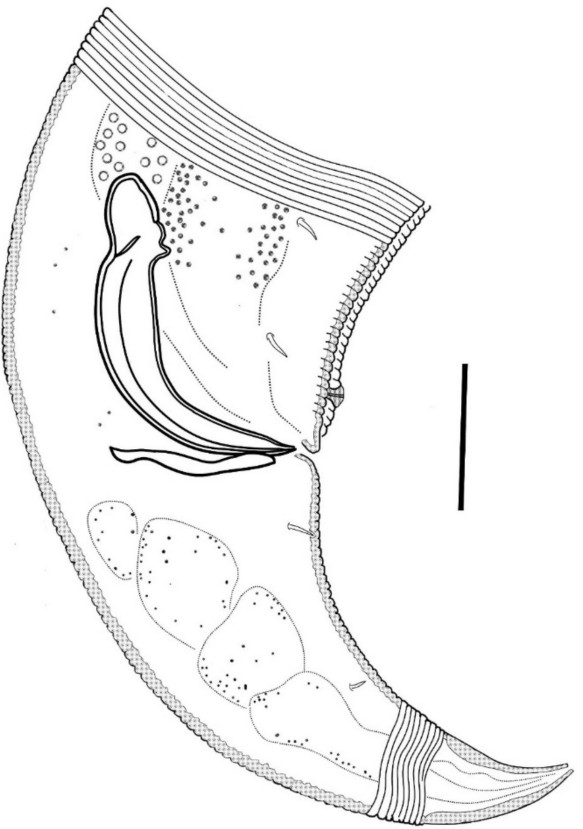

**Figure 9.** *Acanthopharynx micans,* male tail (sample PA13, sponge *Cinachrya* sp.). Scale bar 20 μm.

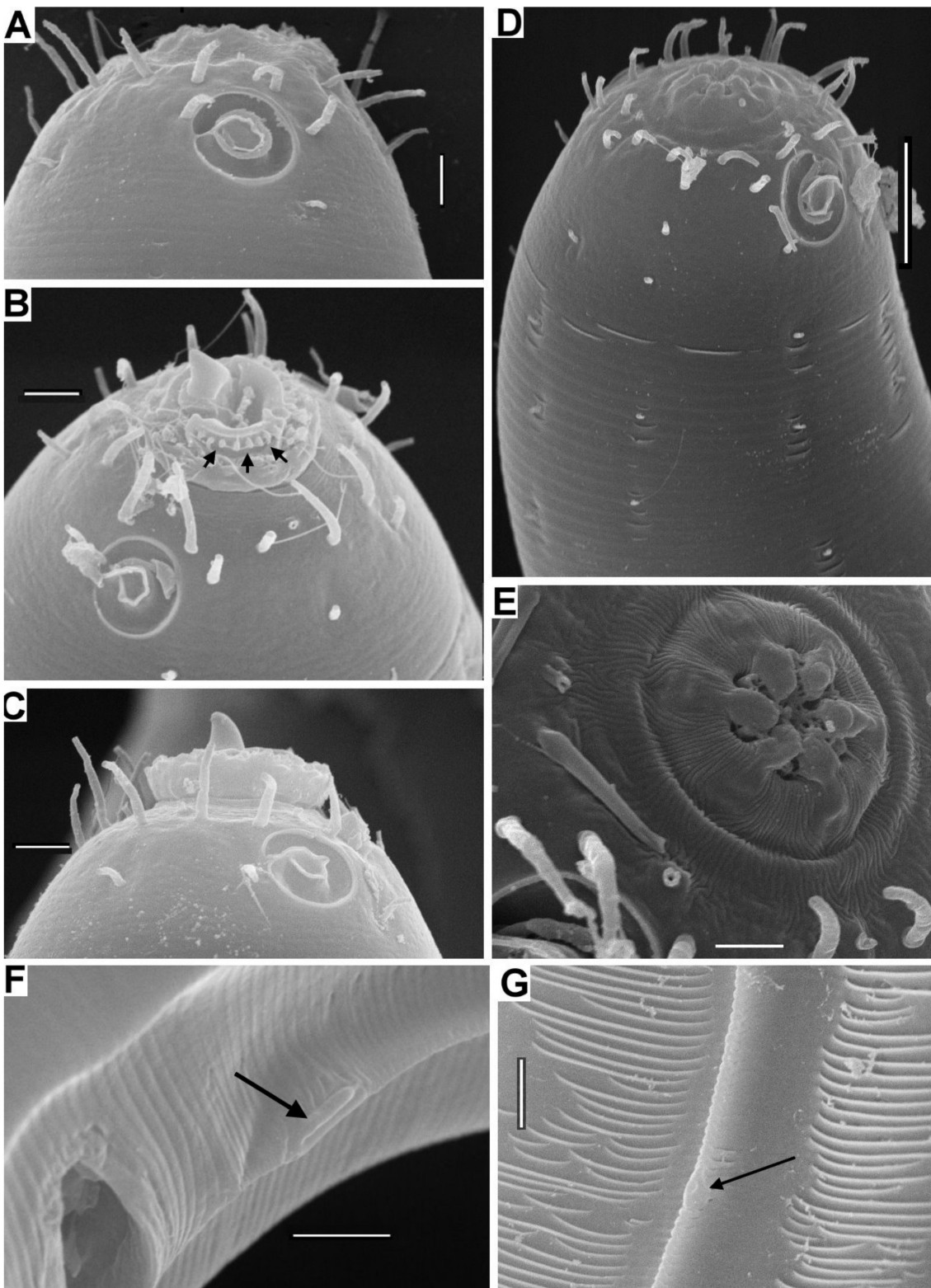

**Figure 10.** *Acanthopharynx micans*, SEM-pictures: (**A**) lateral view of the head; (**B**) head, subapical view; dorsal tooth protruded, little arrows indicate protruded lateroventral row of denticles; (**C**) lateral view of the head, labial region, and dorsal tooth protruded; (**D**) anterior end, laterodorsal view; (**E**) labial region; (**F**) cloacal opening and preanal supplementary papilla (arrow); (**G**) pore on the precloacal ridge (arrow). Scale bars: (**A**) 3 μm, (**B**) 3 μm, (**C**) 3 μm, (**D**) 10 μm, (**E**) 2 μm, (**F**) 3 μm, (**G**) 3 μm.

**Table 4.** Morphometrics of males *Acanthopharynx micans* from samples PA13 and PA21 united.

| Character | Males | | | | |
|---|---|---|---|---|---|
| | *n* | Min–Max | Mean | SD | CV |
| Body length | 41 | 1569–2325 | 1921 | 170.5 | 8.88 |
| Pharynx length | 41 | 292–406 | 332 | 20.8 | 6.28 |
| Tail length | 41 | 84.2–110 | 97.1 | 5.84 | 6.01 |
| a | 41 | 32.3–51.6 | 42.1 | 4.14 | 9.84 |
| b | 41 | 4.65–6.84 | 5.79 | 0.49 | 8.46 |
| c | 41 | 16.4–24.5 | 19.8 | 2.01 | 10.1 |
| c′ | 36 | 2.27–3.16 | 2.61 | 0.19 | 7.28 |
| Body diameter at level of cephalic setae | 40 | 18.9–24.3 | 21.0 | 1.39 | 6.61 |
| Body diameter at level of amphid | 40 | 22.0–30.7 | 26.5 | 1.88 | 7.09 |
| Body diameter at level of nerve ring | 38 | 40.5–52.8 | 44.7 | 3.11 | 6.96 |
| Body diameter at level of cardia | 39 | 38.4–56.0 | 45.1 | 3.73 | 8.26 |
| Body diameter at level of midbody | 41 | 40.2–53.7 | 45.8 | 3.55 | 7.75 |
| Body diameter at level of cloaca | 36 | 33.4–45.0 | 37.2 | 2.45 | 6.58 |
| Body diameter at amphid/Body diameter at cardia, % | 39 | 49.3–73.4 | 59.6 | 4.67 | 7.84 |
| Cephalic capsule length | 41 | 19.9–27.8 | 23.3 | 1.72 | 7.39 |
| Cephalic capsule basal diameter | 41 | 31.2–37.8 | 33.3 | 1.61 | 4.83 |
| Cephalic capsule length/Cephalic capsule basal diameter | 41 | 0.59–0.84 | 0.70 | 0.05 | 7.14 |
| Cephalic setae length | 34 | 7.10–10.9 | 9.00 | 0.09 | 10.0 |
| Amphid width | 40.0 | 6.90–9.90 | 8.30 | 0.85 | 10.2 |
| Distance head apex—amphid | 37 | 3.00–11.1 | 5.55 | 1.72 | 31.0 |
| Stoma maximal width | 41 | 6.80–10.5 | 8.30 | 0.71 | 8.55 |
| Stoma length | 41 | 23.2–50.0 | 31.7 | 4.10 | 13.0 |
| Terminal bulb length | 40 | 161–242 | 202 | 15.9 | 7.88 |
| Terminal bulb length/Total pharynx length, % | 37.0 | 52.0–72.6 | 61.2 | 3.87 | 6.33 |
| Prebulbar pharynx diameter | 37 | 13.3–19.4 | 15.8 | 1.38 | 8.72 |
| Terminal bulb diameter | 41 | 21.3–32.1 | 27.0 | 2.17 | 8.05 |
| Prebulbar pharynx diameter/Terminal bulb diameter, % | 37 | 45.5–79.8 | 59.1 | 6.85 | 11.6 |
| Tail terminal cone length | 38 | 18.8–29.4 | 22.4 | 2.05 | 9.17 |
| Tail terminal cone basal diameter | 38 | 8.80–14.7 | 12.9 | 1.11 | 8.62 |
| Terminal cone length/Total tail length, % | 38 | 19.8–30.3 | 23.3 | 2.29 | 9.83 |
| Terminal cone length/Terminal cone basal diameter | 38 | 1.37–2.31 | 1.75 | 0.21 | 12.0 |
| Spicule arc | 41 | 48.4–74.0 | 61.9 | 5.06 | 8.18 |
| Spicule chord | 41 | 40.8–57.6 | 47.6 | 3.42 | 7.19 |
| Gubernaculum | 40 | 22.0–30.7 | 26.8 | 2.41 | 9.00 |
| Distance cloaca—posteriormost supplementary papilla | 37 | 9.00–15.0 | 11.9 | 1.35 | 11.4 |

All direct measurements are in μm.

**Table 5.** Morphometrics of females *Acanthopharynx micans* from samples PA13 and PA21 united.

| Character | Females | | | | |
|---|---|---|---|---|---|
| | *n* | Min–Max | Mean | SD | CV |
| Body length | 28 | 1590–2269 | 1858 | 132 | 7.41 |
| Pharynx length | 28 | 303–401 | 333 | 23.1 | 6.93 |
| Tail length | 29 | 87.7–115 | 97.7 | 6.03 | 6.18 |
| Distance head apex–vulva | 28 | 846–1125 | 961 | 61.8 | 6.43 |
| a | 27 | 30.2–43.9 | 36.9 | 3.25 | 8.81 |
| b | 26 | 4.91–6.18 | 5.55 | 0.35 | 6.31 |
| c | 27 | 16.1–21.1 | 19.0 | 1.26 | 6.62 |
| c′ | 28 | 2.69–4.28 | 3.41 | 0.33 | 9.68 |
| V, % | 27 | 47.3–54.8 | 51.8 | 1.66 | 3.20 |
| Body diameter at level of cephalic setae | 28 | 19.5–28.5 | 21.4 | 1.73 | 8.10 |
| Body diameter at level of amphid | 25 | 23.3–30.0 | 26.1 | 1.75 | 6.72 |
| Body diameter at level of nerve ring | 26 | 39.9–58.0 | 44.2 | 4.35 | 9.83 |
| Body diameter at level of cardia | 24 | 40.5–61.0 | 45.9 | 4.78 | 10.4 |
| Body diameter at level of midbody | 28 | 43.4–60.0 | 50.5 | 3.99 | 7.90 |

**Table 5.** *Cont.*

| Character | Females | | | | |
|---|---|---|---|---|---|
| | *n* | Min–Max | Mean | SD | CV |
| Body diameter at level of anus | 28 | 21.1–36.0 | 28.9 | 2.65 | 9.18 |
| Body diameter at amphid/Body diameter at cardia, % | 21 | 52.5–65.1 | 57.8 | 3.40 | 5.89 |
| Cephalic capsule length | 28 | 18.6–25.8 | 22.2 | 1.79 | 8.06 |
| Cephalic capsule basal diameter | 28 | 31.0–41.5 | 33.4 | 2.39 | 7.15 |
| Cephalic capsule length/Cephalic capsule basal diameter | 28 | 0.53–0.81 | 0.67 | 0.07 | 10.4 |
| Cephalic setae length | 20 | 7.20–10.8 | 8.99 | 0.93 | 10.3 |
| Amphid width | 23 | 5.90–9.00 | 7.45 | 0.97 | 13.0 |
| Distance head apex—amphid | 23 | 2.70–9.50 | 5.17 | 1.63 | 31.5 |
| Stoma maximal width | 28 | 6.40–10.7 | 8.64 | 1.05 | 12.2 |
| Stoma length | 28 | 25.0–37.0 | 30.8 | 2.40 | 7.80 |
| Terminal bulb length | 26 | 168–244 | 204 | 17.0 | 8.33 |
| Terminal bulb length/Total pharynx length, % | 26 | 52.0–66.5 | 60.7 | 3.09 | 5.09 |
| Prebulbar pharynx diameter | 24 | 13.7–27.3 | 16.5 | 2.68 | 16.2 |
| Terminal bulb diameter | 26 | 25.0–35.7 | 28.3 | 2.76 | 9.77 |
| Prebulbar pharynx diameter/terminal bulb diameter, % | 22 | 45.2–87.2 | 57.5 | 8.60 | 14.9 |
| Tail terminal cone length | 10 | 22.5–32.8 | 26.8 | 3.11 | 11.6 |
| Tail terminal cone basal diameter | 10 | 13.0–18.6 | 15.4 | 1.78 | 11.6 |
| Terminal cone length/Total tail length, % | 28 | 20.9–32.5 | 27.7 | 2.89 | 10.4 |
| Terminal cone length/Terminal cone basal diameter | 28 | 1.44–2.27 | 1.88 | 0.20 | 10.6 |

All direct measurements are in µm.

Material Examined

Forty-one males and twenty-eight females in permanent glycerin slides have been observed, measured, partly pictured, and drawn. Some slides (no catalog numbers) are deposited in the nematode collection of the Center of Parasitology, A.N. Severtsov Institute of Ecology and Evolution of the Russian Academy of Sciences, Moscow, Russia.

Locality

South coast of Cuba in the vicinity of Cienfuegos city, Ancón Beach, 21°71′01.53″–21°75′31.79″ N and 79°99′39.96″–80°02′75.64″ W, depths 9–16 m, sponges, 17–18 November 2019.

Description

Body slender, cylindrical. The cuticle is finely but distinctly annulated; the annulation is uniform along the entire body from the cephalic capsule to the tail terminal cone. In a male, there are 13 annules per 20 µm just posterior to the cephalic capsule, 21–22 annules per 20 µm in the midbody, 20 annules per 20 µm just anterior to the cloaca, 21 annules per 20 µm just anterior to the tail terminal cone.

The cephalic capsule is shaped as a rounded truncate cone. The cuticle of the cephalic capsule is smooth. The labial region is distinctly bordered by a circular furrow. The mouth opening is surrounded by six small lips, which are raised up in some of the specimens. No sensilla were observed in the labial region. There are six papillae (evidently, outer labial sensilla) just posterior to the labial region. For all of the setae, altogether up to 16 or 22 in number, located apically and subapically on the head, are all nearly equal in length and directed forward. The cephalic setae in the lateromedial position are distinguished from other apical setae neither by length nor width. There are two additional sublateral setae between the cephalic setae on either lateral side of the head. In addition to the apical and subapical setae, other more posterior setae on the cephalic capsule are much smaller and sparse, and they are arranged in about ten loose longitudinal rows. The rows of short setae or papillae continue posteriad along the body. Amphideal fovea medium-sized, situated anteriorly on the cephalic capsule, spiral in one turn with a central spot, round in outline.

In the optical section, the somatic cuticle around the cheilostoma light homogeneous is distinctly separated from the main part of the cephalic capsule. The walls of the cheilostoma are complicated with short longitudinal rugae, evidently, there are twelve in number. The lower layer of the apical cheilostomatal cuticle around the mouth differentiated into

cuneiform light-refracted structures. The cuticle of the cephalic capsule looks dense and non-vacuolated, which is thickened considerably to the posterior edge. Pharyngostoma consists of two parts, anterior about cup-shaped gymnostoma and elongated tightly conoid stegostoma, both with cuticularized walls. There is a prominent solid fang-shaped dorsal tooth at the anterior edge of stegostoma. Two lateroventral comb-like rows of minute denticles are opposed to the dorsal tooth. Both dorsal teeth and rows of denticles can push out a bit. The denticles, however, are very fine and hardly discernible in many specimens. The entire stegostoma are elongated and narrowing posteriad is evident, and its posterior end is marked with a notch. The pharynx is evenly muscular throughout its length, with a very distinct thick internal cuticular lining. The posterior portion over half of the entire pharynx constitutes a long bulb-like widening, which is distinctly separated from the narrow anterior part; the internal cuticular lining is also enlarged in the posterior widening of the pharynx. A hardly visible nerve ring is located at the posterior end of the narrow anterior part of the pharynx. The cardia is obscure.

No ventral gland is visible.

The male reproductive system is monorchic, and the testis are situated to the right of the intestine. The spermatozoa are irregularly ovoid, with granular content. Spicules paired, equal, arcuate, with anterior moderately differentiated proximal knobs and pointed distal tips. The gubernaculum consists of paired stick-like bars perpendicular to the longitudinal body axis. There is a long midventral precloacal shallow furrow extending anteriad from the cloaca ridge. A low midventral ridge rising from the furrow bears a row of hardly discernible supplementary pores and a posteriormost supplementary organ with the appearance of a soft wart close to the cloaca.

Tail conical, pointed, slightly curved ventrally, with few small pre- and postanal subventral setae. Terminal cone with smooth cuticle. No sensilla was found on the tail and in the precloacal area.

Remarks

*Acanthopharynx micans* is apparently the most common and widespread species of the genus: formerly, *A. micans* was recorded in several localities of the Mediterranean, Red Sea, and Maldive Islands [27]. The species was found only on shallows and not so much in sediments as on corals, algae, and mussel banks [33,35,45,49]. The cuban specimens do not differ from those of other regions in structural details. However, they are distinguished by larger body size (Table 6), which may relate to their habitation in sponges under the enhanced feeding condition.

Another *Acanthopharynx* species previously known in nearly the same location is *A. denticulata* [31], but this species has been found in bottom sediments, not in sponges. Morphometrically, Cuban *A. micans* conforms with the sympatric *A. denticulata* population in body length and body ratios but differs by a longer pharynx (292–406 versus 214–286 µm), pharynx shape with ellipsoid versus cylindrical posterior widening and precloacal midventral row of one prominent posteriormost papilla with a number of supplementary pores versus 13–16 equal supplementary pits.

*Acanthopharynx parva* sp. n.

Figures 11–13, Tables 7 and 8.

**Table 6.** Morphometrical comparison of males of valid *Acanthopharynx* species (data rounded, names and sources of original diagnoses in bold). "?" means no data.

| Species | Body Length | a | b | c | c′ | Spicule Length | Source |
|---|---|---|---|---|---|---|---|
| *affinis* | 2150 | 27 calc | ? | 25 calc | ? | ? | **[26]** |
| *affinis* | 2208 | 42 | 7.2 | 35 | 2 | 1.4 cbd | [33] |
| *denticulata* | 2170–2780 | 44–49 | 8.1–9.7 | 23–31 | 2.0–2.5 | 65 (1.3 cbd) | **[29]** |
| *denticulata* | 1234–1745 | 24–43 | 5–7 | 16–21 | 1.6–2.2 | 64–81 | [31] |
| *distechei* | 1750 | 19.6 | 4.8 | 13.8 | 2.3 calc | 117 | **[30]** |
| *dormitata* | 1809–2311 | 38–50 | 6–8 | 22–29 | 1.6–1.8 | 59–63 | **[8]** |
| *japonica* | 2470–2850 | 31–44 | 6.2–7.7 | 20–23 | 1.69 | ? | **[37]** |
| *micans* | 1250 | ? | 4 calc | ? | 2 calc | ? | **[42]** |
| *micans* | 1980 | 47 | 6.9 | 26 | 2.2 | ? | [43] |
| *micans* | 1932 | 40 | 6.5 | 27 | 2.4 | 1.2 cbd | [45] |
| *micans* | 1480-1770 | 31–32 | 6.2–6.3 | 20–23 | 2 | 49 | [44] |
| *micans* | 1418 | 27 | 5 | 16 | 2 | 55 | [35] |
| *micans* | 1569–2325 | 32–52 | 4.6–6.9 | 16–25 | 2.2–3.2 | 48–74 | present paper |
| *micramphis* | (juv) 1036 | (juv) 23.5 | (juv) 3.7 | (juv) 17.2 | 1.75 | ? | **[33]** |
| *nuda* | >2000 | 23 calc | 4.85 calc | 13 calc | 2.66 calc | ? | **[46]** |
| *parva* **sp. n.** | 1024–1370 | 26–42 | 5.7–7.8 | 10.6–14.4 | 3.2–3.9 | 37–75 | **present paper** |
| *perarmata* | 2000 | 24 calc | ? | 29 calc | 1.7–1.8 calc | ? | **[26]** |
| *perarmata* | ? | ? | ? | ? | 1.8 | ? | [45] |
| *rigida* | 2828 | 50 | 6.8 | 24 | 3.1 | 1.3 cbd | **[45]** |

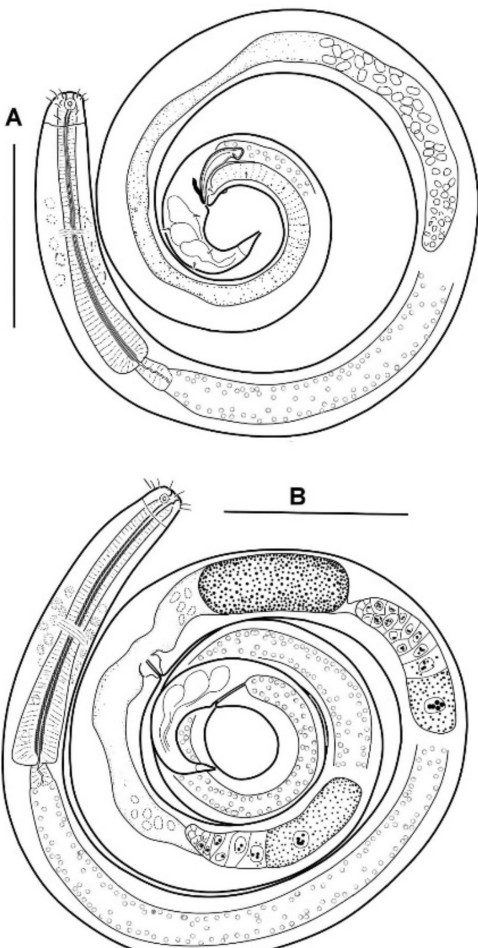

**Figure 11.** *Acanthopharynx parva* sp. n., entire: (**A**) holotype male; (**B**) allotype female. Scale bars 100 μm.

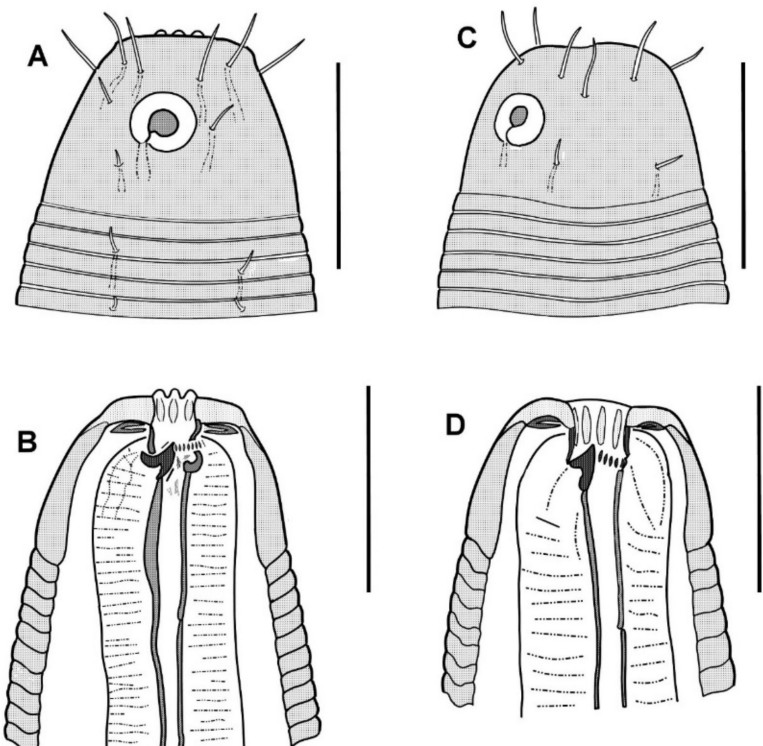

**Figure 12.** *Acanthopharynx parva* sp. n., anterior ends: (**A**) holotype male, surface view; (**B**) holotype male, optical section; (**C**)—allotype female, surface view; (**D**) allotype female, optical section. Scale bars 20 μm.

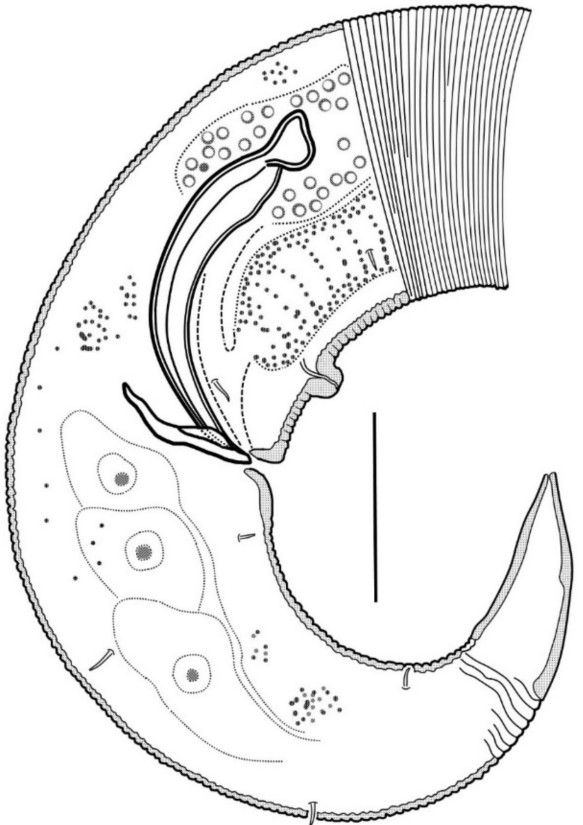

**Figure 13.** *Acanthopharynx parva* sp. n., holotype male tail. Scale bar 20 μm.

**Table 7.** Morphometrics of males *Acanthopharynx parva* sp. n. from samples PA13 and PA21 united.

| Character | Holotype Male | Holotype and Paratype Males | | | | |
|---|---|---|---|---|---|---|
| | | *n* | Min–Max | Mean | SD | CV |
| Body length | 1058 | 12 | 1024–1370 | 1194 | 102 | 8.38 |
| Pharynx length | 168 | 12 | 156–177 | 167 | 7.97 | 4.83 |
| Tail length | 99.0 | 12 | 84.0–99.6 | 92.4 | 5.58 | 5.87 |
| a | 28.4 | 12 | 26.8–42.0 | 33.9 | 4.58 | 13.5 |
| b | 6.30 | 12 | 5.79–7.78 | 7.15 | 0.67 | 8.47 |
| c | 10.6 | 12 | 10.6–14.4 | 12.9 | 1.13 | 8.96 |
| c′ | 3.76 | 10 | 3.27–3.86 | 3.56 | 0.18 | 4.79 |
| Body diameter at level of cephalic setae | 17.8 | 12 | 15.5–18.5 | 16.9 | 1.04 | 5.92 |
| Body diameter at level of amphid | 21.5 | 12 | 20.6–29.5 | 23.0 | 2.30 | 8.88 |
| Body diameter at level of nerve ring | 37.8 | 12 | 31.3–45.0 | 35.2 | 3.69 | 9.00 |
| Body diameter at level of cardia | 36.5 | 10 | 33.5–45.0 | 36.0 | 3.52 | 8.40 |
| Body diameter at level of midbody | 37.2 | 12 | 30.9–44.8 | 35.3 | 3.89 | 10.0 |
| Body diameter at level of cloaca | 26.5 | 10 | 25.6–27.1 | 26.4 | 0.49 | 1.86 |
| Body diameter at amphid/Body diameter at cardia, % | 58.9 | 10 | 54.6–67.0 | 63.6 | 3.94 | 6.78 |
| Cephalic capsule length | 16.7 | 12 | 15.0–21.1 | 17.1 | 1.67 | 8.65 |
| Cephalic capsule basal diameter | 28.5 | 12 | 24.6–32.9 | 26.6 | 2.12 | 6.67 |
| Cephalic capsule length/Cephalic capsule basal diameter | 0.65 | 12 | 0.57–0.71 | 0.64 | 0.04 | 4.69 |
| Cephalic setae length | 6.6 | 12 | 4.0–6.6 | 5.32 | 0.80 | 15.5 |
| Cephalic setae length in % cbd | 37.1 | 9 | 25.6–37.1 | 32.1 | 4.15 | 12.9 |
| Amphid width | 6.6 | 12 | 5.5–10.5 | 7.07 | 1.37 | 16.9 |
| Amphid width in % cbd | 30.7 | 12 | 24.0–37.1 | 30.7 | 4.48 | 14.6 |
| Distance head apex—amphid | 5.0 | 12 | 3.7–9.1 | 5.72 | 1.63 | 28.6 |
| Stoma maximal width | 5.1 | 11 | 4.6–6.4 | 5.13 | 0.55 | 9.98 |
| Stoma length | 21.8 | 12 | 18.0–23.4 | 20.9 | 1.74 | 8.21 |
| Posterior pharynx widening length | 61.0 | 8 | 40.0–86.4 | 61.0 | 15.5 | 31.5 |
| Posterior pharynx widening length in % of total pharynx length | 36.3 | 8 | 23.2–48.4 | 35.8 | 8.23 | 19.3 |
| Pharynx diameter at nerve ring | 14.3 | 12 | 13.0–14.8 | 14.1 | 0.51 | 3.34 |
| Pharynx diameter at posterior widening end | 29.4 | 11 | 21.0–29.4 | 26.1 | 2.48 | 10.2 |
| Pharynx diameter at nerve ring in % of posterior pharynx end diameter | 48.6 | 11 | 48.4–69.0 | 54.6 | 6.05 | 11.1 |
| Tail terminal cone length | 23.4 | 12 | 18.4–29.0 | 22.4 | 3.01 | 12.6 |
| Tail terminal cone basal diameter | 10.5 | 11 | 9.4–15.0 | 11.2 | 1.61 | 12.7 |
| Terminal cone length in % entire tail length | 23.5 | 12 | 19.8–34.5 | 24.3 | 3.86 | 14.4 |
| Terminal cone length/Terminal cone basal diameter | 2.23 | 11 | 1.47–2.28 | 2.03 | 0.26 | 13.4 |
| Spicule arc | 44.8 | 12 | 37.4–75.0 | 45.4 | 9.82 | 18.6 |
| Spicule chord | 36.4 | 12 | 29.6–51.0 | 35.3 | 5.32 | 12.7 |
| Spicule chord/cloacal body diameter | 1.37 | 10 | 1.16–1.38 | 1.28 | 0.07 | 5.47 |
| Gubernaculum length | 17.0 | 12 | 14.5–32.0 | 18.6 | 4.44 | 20.7 |
| Distance cloaca—posteriormost supplementary papilla | 12.0 | 12 | 8.00–19.5 | 12.6 | 2.93 | 21.6 |

**Table 8.** Morphometrics of females *Acanthopharynx parva* sp. n. from samples PA13 and PA21 united.

| Character | Allotype Female | Allotype and Paratype Females | | | | |
|---|---|---|---|---|---|---|
| | | *n* | Min–Max | Mean | SD | CV |
| Body length | 1422 | 13 | 1220–1435 | 1312 | 67.7 | 5.16 |
| Pharynx length | 181 | 13 | 166–195 | 178 | 7.33 | 4.12 |
| Tail length | 82.2 | 13 | 82.2–107 | 92.6 | 6.48 | 7.00 |
| Distance head apex–vulva | 778 | 13 | 607–778 | 668 | 48.6 | 7.27 |
| a | 34.4 | 13 | 25.9–34.4 | 29.5 | 2.31 | 7.83 |
| b | 7.86 | 13 | 6.79–8.40 | 7.38 | 0.47 | 6.37 |
| c | 17.3 | 13 | 12.7–17.3 | 14.2 | 1.30 | 9.14 |
| c′ | 2.98 | 12 | 2.98–4.53 | 3.82 | 0.41 | 10.7 |
| V, % | 54.7 | 12 | 49.2–55.1 | 51.2 | 1.96 | 3.83 |

**Table 8.** *Cont.*

| Character | Allotype Female | Allotype and Paratype Females | | | | |
|---|---|---|---|---|---|---|
| | | *n* | Min–Max | Mean | SD | CV |
| Body diameter at level of cephalic setae | 18.0 | 13 | 15.5–19.0 | 17.1 | 1.07 | 6.24 |
| Body diameter at level of amphid | 22.0 | 13 | 20.0–24.2 | 22.1 | 1.37 | 6.19 |
| Body diameter at level of nerve ring | 37.0 | 3 | 35.1–38.9 | 36.6 | 2.00 | 5.46 |
| Body diameter at level of cardia | 36.9 | 10 | 33.7–40.0 | 37.5 | 1.76 | 4.69 |
| Body diameter at level of midbody | 41.3 | 13 | 39.4–48.3 | 44.6 | 2.83 | 6.34 |
| Body diameter at level of anus | 27.6 | 12 | 22.4–27.9 | 24.3 | 1.74 | 7.16 |
| Body diameter at amphid/Body diameter at cardia, % | 59.6 | 10 | 52.4–63.4 | 58.4 | 3.07 | 5.26 |
| Cephalic capsule length | 15.7 | 13 | 13.9–18.3 | 15.7 | 1.13 | 7.18 |
| Cephalic capsule basal diameter | 25.9 | 13 | 25.2–27.6 | 26.6 | 0.69 | 2.59 |
| Cephalic capsule length/Cephalic capsule basal diameter | 0.61 | 11 | 0.52–0.69 | 0.59 | 0.04 | 6.78 |
| Cephalic setae length | ? | 8 | 4.30–7.50 | 5.89 | 1.05 | 17.8 |
| Amphid width | 4.50 | 12 | 4.50–6.80 | 5.56 | 0.58 | 10.4 |
| Distance head apex—amphid | 6.00 | 12 | 4.30–8.60 | 5.85 | 1.23 | 21.3 |
| Stoma maximal width | 4.50 | 13 | 4.50–6.30 | 5.20 | 0.51 | 9.81 |
| Stoma length | 20.0 | 13 | 20.0–24.4 | 21.4 | 1.31 | 6.13 |
| Terminal bulb length | 71.9 | 12 | 71.5–88.2 | 78.6 | 5.58 | 7.10 |
| Terminal bulb length/Total pharynx length, % | 39.7 | 12 | 39.7–53.0 | 45.1 | 3.58 | 7.93 |
| Midpharynx diameter at nerve ring | 14.5 | 12 | 12.9–15.2 | 14.1 | 0.75 | 5.31 |
| Posterior pharynx diameter | 27.6 | 12 | 24.6–32.0 | 27.8 | 1.75 | 6.29 |
| Midpharynx diameter/Posterior pharynx diameter, % | 52.5 | 12 | 41.9–57.7 | 50.9 | 4.26 | 8.36 |
| Tail terminal cone length | 22.0 | 13 | 21.7–26.0 | 23.7 | 1.33 | 5.62 |
| Tail terminal cone basal diameter | 12.1 | 13 | 10.4–13.0 | 11.3 | 0.74 | 6.54 |
| Terminal cone length/Total tail length, % | 26.8 | 13 | 22.1–28.3 | 25.7 | 1.99 | 7.76 |
| Terminal cone length/Terminal cone basal diameter | 1.82 | 13 | 1.82–2.37 | 2.10 | 0.15 | 7.14 |

Etymology

The species name (from the Latin "parvus," little) reflects the smaller body sizes than the cohabiting *Acanthopharynx micans*.

Material Examined

All types of specimens are mounted in permanent glycerin slides. Holotype male (slide 50/12) and allotype female (slide 50/13) were deposited in the nematode collection of the Center of Parasitology, A.N. Severtsov Institute of Ecology and Evolution of the Russian Academy of Sciences, Moscow, Russia. Twelve male and thirteen female paratypes are deposited in the same collection.

Type Locality

The south coast of Cuba in the vicinity of Cienfuegos city, Ancón Beach, 21°71′01.53″–21°75′31.79″ N and 79°99′39.96″–80°02′75.64″ W, depths 9–16 m, sponges, 17–18 November 2019.

Description

The body is cylindrical with a rounded head end and a short conical tail. The cuticle is faint but distinctly annulated except for the cephalic helmet (cephalic capsule) and terminal tail cone. In the holotype male, there are 13 annules within 20 μm just behind the cephalic capsule, 25 annules within 20 μm just behind the cardia, 30 annules within 20 μm in the midbody, 22 annules within 20 μm in the midtail (dorsal convex side). In a paratype female, there are 12 annules within 20 μm just behind the cephalic capsule, and 20 annules within 20 μm in the midbody.

The cephalic capsule, which is non-annulated, is formed by a dense somatic cuticle. The cuticle of the cephalic capsule is light-refracted, non-sculptured, and non-vesiculated, having a thickened posteriad. There are about twelve short setae forming a subapical crown; four cephalic setae do not differ in size from other setae of the crown; the cephalic setae are identified based on their lateroventral position. Posterior to the subapical crown, the other setae on the cephalic capsule are minute and arranged irregularly posterior to the

subapical crown. The transcuticular nerve canals of setae and amphid are very distinctly visible. There are sparse rows of tiny somatic setae extended along the body.

The amphideal fovea presents a circular interrupted ring with a cuticular spot in the center; actually, the fovea is a spiral coiled ventrally in one turn.

The mouth opening is surrounded by six small lips. The cheilostoma features levigate longitudinal rugae. The anterior part of the pharyngostoma features cuticularized walls, teeth, and denticles. The dorsal tooth is solid, claw-like, and movable; it is opposed by a pair, left and right lateroventral flanges of tiny denticles, also protrusible. The posterior part of the pharyngostoma is elongated and tight; its posterior ending is marked by a light flexure. There are cuniform solid cuticular structures on the internal surface of the circumoral apical cuticle; they are connected with longitudinal muscles and likely serve by opening the mouth.

Pharynx with the finest transversal muscular striation throughout its length. The posterior fourth to third of the pharynx is enlarged and in the shape of an elongated isosceles triangle. The internal cuticular lining of the pharynx is widened, especially in the posterior half. The cardia is external and narrow.

The male reproductive system is monorchic and situated to the right of the intestine. The spermatozoa is ovoid and have a granular content; they are about $8 \times 4$ μm in size. The paired spicules are equal, arcuate, distally pointed, and proximally knobbed. The gubernaculum presents as paired bars oriented dorso-ventrally. An only evident precloacal supplementary papilla is situated midventrally close to the cloacal opening. Precloacal midventral papillae or pores are not visible.

In females, the ovaries are antidromously reflected, anterior left and posterior right to the midgut.

Tail short conical, with three poorly discernible caudal gland cell bodies within. Cuticle of the tail hind part non-annulated, thus shaping a terminal cone.

Diagnosis

*Acanthopharynx*. Body length 1020–1440 μm, a 25–42, b 5.8–8.4, c 10.6–17.3, c' 3–4.5. Cephalic setae 4–7.5 μm long. The amphideal fovea are spirally coiled in one turn and circular in outline. The buccal armature consists of a movable dorsal tooth and two transversal lateroventral rows of minute denticles. The pharynx is short and continually widened to the cardia without a sharply defined bulb. The spicules are 29–75 μm long. The precloacal midventral organs are only present with papilla. There were no distinct postcloacal supplementary organs.

Differential diagnosis

*Acanthopharynx parva* sp. n. differs from all the other *Acanthopharynx* species (with possible exception of *A. denticulata*) due to the shape of the pharynx gradually widening to the cardia and lacking a conspicuous elongate swelling or bulb. In addition to that, *A. parva* clearly differs from those *Acanthopharynx* species characterized by unique prominent features. Thus, *A. parva* differs from *A. affinis* by the presence of the prominent precloacal papilla; from *A. denticulata* by the absence of subventral teeth in the stoma (*A. denticulata* possesses smaller subventral teeth in addition to a large dorsal tooth and two rows of minute denticles) and the presence of prominent precloacal papilla; from *A. distechei* by longer cephalic setae (4–7.5 μm versus 3–3.5 μm), round versus longitudinally oval amphideal fovea, short spicules 29–75 versus 117 μm, oviparity versus viviparity; from *A. dormitata* by a discrepant set of precloacal supplementary organs (one prominent posteriormost papilla versus nine precloacal and two postcloacal papillae); from *A. micramphis* by presence of the preanal papilla (no supplementary organs not observed on *A. micramphis*), from *A. nuda* by another set of supplementary organs (an only conspicuous precloacal papilla versus six precloacal papillae along a distance equal to the tail length); from *A. perarmata* also by supplementary organs (an only precloacal papilla versus three almost equidistant papillae).

The differences in *Acanthopharynx parva* from *A. japonica* and *A. rigida* are somewhat less evident since those species were described seventy or more years ago. Therefore, some

structures, foremost the buccal armature and the pattern of the supplementary papillae need to be specified in detail. Morphometric differences of *A. parva* from *A. japonica*, *A. micans*, *A. rigida* and other species are summarized in Table 6.

In addition, it is necessary to consider the relations of *A. parva* to two cohabitated *Acanthopharynx* species populations. *A. parva* differs from *A. micans* (present paper) in the shape of the pharynx (gradually broadening to the posterior end versus distinctly separated elongate widening) and lesser body length (1020–1440 µm versus 1569–2325 µm), pharynx length (156–195 µm versus 292–406 µm), c of males (10–15 versus 16–25). *A. parva* differs from sympatric *A. denticulata* [31] by pharynx shape (posterior widening ellipsoid in *A. denticulata*), index c of males (10–15 versus 16–21), c' of males (3.2–3.9 versus 1.6–2.2), singular prominent precloacal papilla versus 13–16 equal supplementary pits.

### 3.4. Distribution of Nematode Species among Sponges

Four of the 25 sponge samples (*Ircinia felix*, *Aplysina fulva*, *Ircinia* sp., indet sp.) contained no nematode specimens. The highest number of nematode specimens were found in the *Cinachyrella* sp., followed by the unidentified sponge species and *Aiolochroia crassa* (Table 9). Overall, 26 nematode species belonging to 22 genera and 13 families were found in all samples. Prominent dominancy of the family Desmodoridae (50–95% of all individuals) occurs in all the samples with rich nematode populations. Desmodoridae species are followed by Chromadoridae (15–40% of all individuals) (Figure 14). The family Desmodoridae is presented by eight species, whereas the family Chromadoridae—by ten species (Figure 15). The most abundant species also belong to Desmodoridae and Chromadoridae. High proportions of females with fertilized eggs in uteri and high proportions of juveniles suggest for successful reproduction and realization of a complete life cycle within sponges.

**Table 9.** Occurrence of nematodes and *Acanthopharynx* species in sponges.

| Sample Number | Sponge Species (? Means the Species Identification is Rough or Provisional) | Total Number of Nematode Specimens Extracted from the Sample | Total Number of Nematode Species Per Sample | Proportion of *Acanthopharynx* Specimens in % of Total Nematode Number | % Ratio *A. micans:A. parva* |
|---|---|---|---|---|---|
| PA1 | *Ircinia felix* | 0 | 0 | 0 | - |
| PA2 | *Aiolochroia crassa* | 63 | 17 | 26 | 100:0 |
| PA3 | *Aplysina insularis* | 3 | 2 | 0 | - |
| PA4 | *Niphates digitales* | 2 | 1 | 0 | - |
| PA5 | unidentified | 1 | 1 | 0 | - |
| PA6 | *Aplysina fulva* | 0 | 0 | 0 | - |
| PA7 | *Ircinia* sp. | 0 | 0 | 0 | - |
| PA8 | unidentified | 109 | 7 | 16 | 100:0 |
| PA9 | unidentified | 0 | 0 | 0 | - |
| PA10 | *Aiolochroia crassa* (?) | 11 | 6 | 8.3 | 100:0 |
| PA11 | *Verongula rigida* | 1 | 1 | 0 | - |
| PA12 | *Aiolochroia crassa* (?) | 60 | 3 | 47 | - |
| PA13 | *Cinachryrella* sp. | 435 | 10 | 70 | 73:27 |
| PA14 | *Cinachryrella* sp. | 2 | 1 | 0 | - |
| PA15 | *Verongula rigida* | 5 | 3 | 0 | - |
| PA16 | *Verongula rigida* | 4 | 2 | 0 | - |
| PA17 | *Aiolochroia crassa* | 8 | 4 | 0 | - |
| PA18 | *Aiolochroia crassa* | 16 | 7 | 3 | 67:33 |
| PA19 | *Cinachryrella* sp. | 101 | 6 | 23 | 100:0 |
| PA20 | *Verongula rigida* | 1 | 1 | 1 | 100:0 |
| PA21 | *Cinachryrella* sp | 193 | 6 | 76 | 87:13 |
| PA22 | *Aplysina fulva* | 3 | 2 | 66 | 100:0 |
| PA23 | *Iotrochota birotulata* | 3 | 3 | 0 | - |
| PA24 | *Iotrochota birotulata* | 1 | 1 | 0 | - |
| PA25 | unidentified | 19 | 4 | 53 | - |

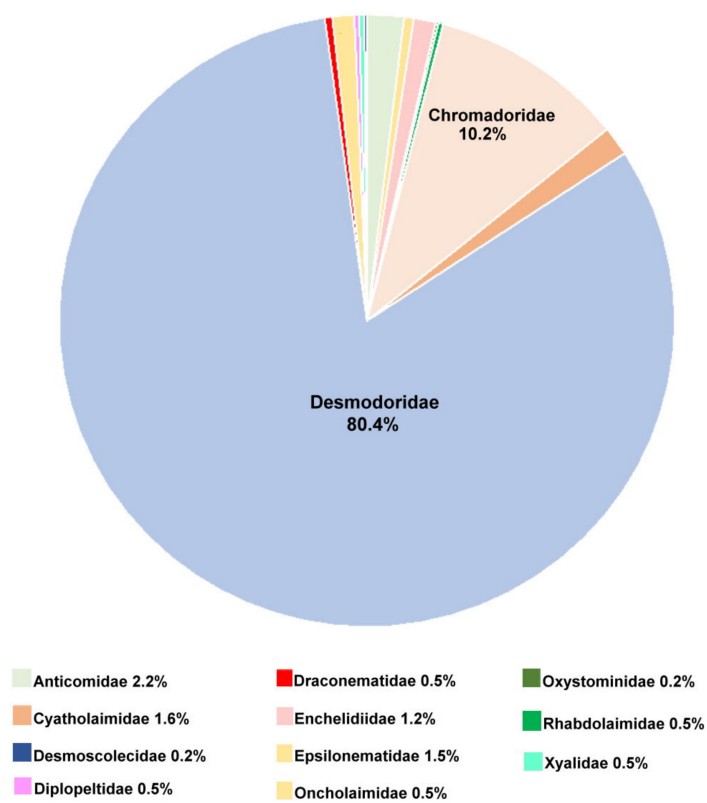

**Figure 14.** The proportion of families (numbers of individuals) in the total nematode population of demosponges.

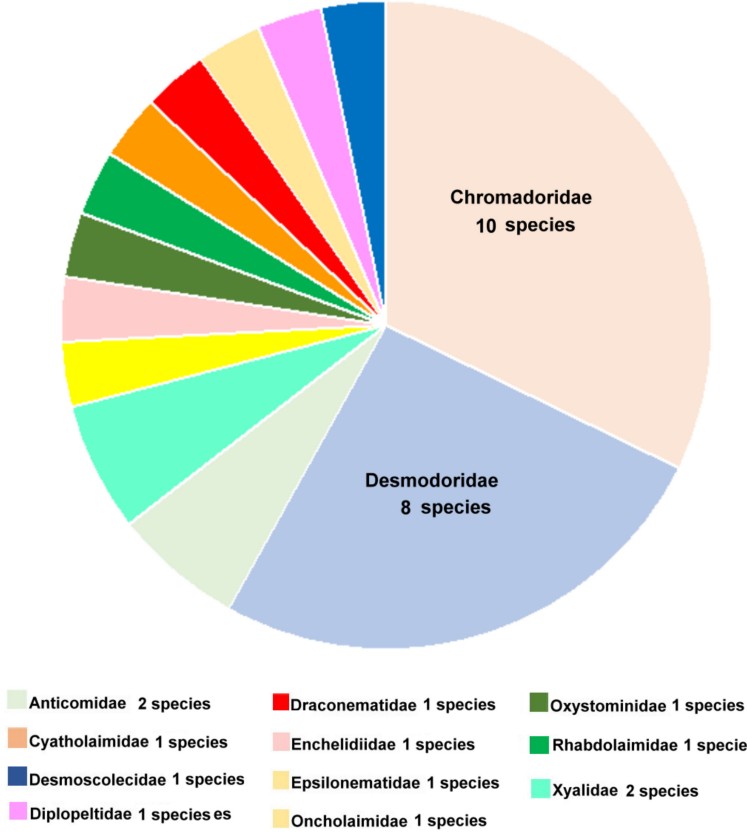

**Figure 15.** Species diversity in families constituting total nematode population in demosponges.

The comparison of the sponge nematode population with nematode communities typical of shallow carbonate coarse sediments, e.g., Refs. [50–52] reveals some features of nematode communities associated with demosponges:

(1)  Generally low diversity with a disproportional predominance of the family Desmodoridae. On a scale of the World Ocean, the proportion of the Desmodoridae family in nematode associations increases in the tropical zone on medium-grain carbonate sediments. However, the proportion of Desmodoridae in the sponges studied is much higher than in adjacent bottom sediments.

(2)  The sponge nematode associations lack species of the family Xyalidae, one of the commonest and most significant taxa in various bottom sediments of the World Ocean. Thus, the nematode population in demosponges does not present an impoverished or random sample from the nematode community of subjacent sediment by a peculiar community.

(3)  There are almost no species defined as deposit feeders (feeding types 1A and 1B of Wieser, 1953 [53]) in the sponge samples. Actually, the diet of the deposit feeders often consists of bacteria. On the contrary, epigrowth feeders (feeding type 2A of Wieser) prevail in sponges. Epigrowth (or epistratum) feeders are a group of various taxa possessing in buccal cavity movable teeth for detaching fungi, unicellular algae, and other protists fastened to a substrate. Thus, epigrowth feeders largely do not ingest bacteria but bigger particles, i.e., eucaryote protists.

*Acanthopharynx*, the most abundant taxon of the nematode community in demosponges, also belongs to Desmodoridae and has the feeding type of epigrowth feeders, according to their complicated movable buccal armature. However, microscopic examination of several tens individuals has not elucidated the diet of the *Acanthopharynx* since all the intestines studied were empty. We suppose that both *Acanthopharynx* species may consume cells and intercellular matrix of the sponge host that do not leave marks in the intestine, such as solid fragments of spicules, collagen, and spongin fibers.

### 4. Conclusions

Some problems considered in this project still need to be solved and pushed off for the future. Thus, we could not identify a third *Acanthopharynx* entity revealed by molecular methods but undistinguished morphologically from *Acanthopharynx micans*. Additionally, the specimens deposited in the GenBank as *A. micans* by other researchers present another clade that differs from known species of the genus. It probably means that the wide distribution of *A. micans* in the literature covers a number of siblings or closely related species.

Other nematode species in sponges, foremost, the species of Desmodoridae, are in store for future studies. The interesting question is, what is the nature of the fine interaction of nematodes and sponges, particularly the food source for nematodes dwelling inside sponges? Additionally, not incurious is the phenomenon of coexistence of two (or more?) closely related species of *Acanthopharynx* in a limited space within the same sponge individual, resources partitioning, and possible ecological niches. The ratio of two *Acanthopharynx* species within sponges may also be a promising issue. In the sponge specimens housing *Acanthopharynx*, *A. micans* either predominates over *A. parva* or presents a sole species—but we suppose that the ratio may overturn under different conditions such as different depths, seasons, sponge species, etc.

**Supplementary Materials:** The following supporting information can be downloaded at: https://www.mdpi.com/article/10.3390/d15010048/s1, Table S1: *Acanthopharynx micans* Morphometrics Sample PA13; Table S2: *Acanthopharynx micans* Morphometrics Sample PA21; Table S3: *Acanthopharynx parva* Morphometrics Sample PA13; Table S4: *Acanthopharynx parva* Morphometrics Sample PA21.

**Author Contributions:** Conceptualization, A.T.; methodology, A.T., P.R.G., U.S. and V.M.; software, A.T., U.S. and V.M.; validation, A.T., U.S. and V.M.; formal analysis, A.T., U.S. and V.M.; investigation, A.T., P.R.G., U.S. and V.M.; resources, A.T., P.R.G., U.S. and V.M.; data curation, A.T., U.S. and P.R.G.; writing—original draft preparation, A.T., U.S. and V.M.; writing—review and editing, A.T., P.R.G., U.S. and V.M.; visualization, A.T., U.S. and V.M.; supervision, A.T.; project administration, A.T.; funding acquisition, A.T. All authors have read and agreed to the published version of the manuscript.

**Funding:** The research was funded by the Russian Foundation of Basic Research, grant number 20-54-56038. APC was funded by the Russian Foundation of Basic Research, grant number 20-54-56038.

**Institutional Review Board Statement:** Not applicable.

**Data Availability Statement:** Not applicable.

**Acknowledgments:** We thank Viatcheslav N. Ivanenko (Department of Invertebrate Zoology, Moscow State University, Russia) and Maickel Armenteros (Instituto de Ciencias del Mar y Limnología, Universidad Nacional Autónoma de México, Mexico) for the suggested concept of the study associations of marine nematodes with invertebrates. We are also grateful to José A. Pérez-García (Centro de Investigaciones Marinas, Universidad de La Habana, Cuba) and V.N. Ivanenko for participation in fieldwork sampling.

**Conflicts of Interest:** The authors declare no conflict of interest.

## Abbreviations

| | |
|---|---|
| a | body length divided by body diameter at midbody |
| b | body length divided by pharynx length |
| c | body length divided by tail length |
| c′ | tail length divided by anal/cloacal diameter |
| calc | calculated or measured from measurements and/or figures |
| cbd | corresponding body diameter |
| CV | coefficient of variation (SD divided by the mean, in %). |
| SD | standard deviation |
| V | distance from anterior and to vulva in % of entire body length |

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
