# Peer review of "Acanthopharynx Marine Nematodes (Nematoda, Chromadoria, Desmodoridae) Dwelling in Tropical Demosponges: Integrative Taxonomy with Description of a New Species†"

_diversity, doi:10.3390/d15010048_

Round 1

Reviewer 1 Report

Excellent well documented and illustrated paper.

The period from lines 29 to 30 seems incomplete.

The period from lines 134 to 163 should be moved to the materials and methods or the introduction

Author Response

Dear Reviewer, thank you for your evaluation and remarks. 

The period from lines 29 to 30 seems incomplete. - Improved (marked with yellow).

The period from lines 134 to 163 should be moved to the materials and methods or the introduction. - I am sorry, I don’t understand the remark. The period from lines 134 to 163 comrises hind part of Material and methods (incl. Abbreviations) and prologue of Results. It is impossible to move this part of the text entirely somewhere. Maybe, this is a misprint, and the Reviewer means the period from lines 154 to 163 (beginning of Morphometric analysis)? If yes, I ask to leave it in place. The point is that the objective of the morphometric analysis is already mentioned briefly in the Introduction while numbers of measured individuals, characters and ratios as well as techniques and algorithms of analyses are mentioned in the Materials and methods. In the Results and discussions, we describe necessity of morphometric analysis for the project as well as some peculiarities for choosing characters in samples.

Reviewer 2 Report

This is well-structured and planned study. The description of the new species, revision of the Acanthopharynx genus and conclusions are valid. All methods were used correctly. Description is supported with beautiful illustrations. The manuscript also provides the valuable information of nematode fauna inhabiting sponges, which have hardly been known so far. This is very interesting manuscript and definitely deserved to be published. However, several changes are suggested for improvement before publication. The biggest issue is the language. Before this paper can be accepted it need significant editing. The authors’ meaning is in some cases difficult to understand. Since English is not my native language, I didn’t make many corrections throughout the MS. Therefore, I’d recommended prior to resubmission ask for help of a colleague or other person proficient in scientific English.  All my other comments and suggestions are included in the pdf file that I have submitted.

Beyond that, I'm looking forward to seeing this manuscript published.

Author Response

Dear Reviewer,

I am very thankful for your remarks and linguistic corrections. I have followed your remarks and improved the text as completely as possible. Improved words and sentences are marked with blue.

Reviewer 3 Report

1) REORDER THE SECTIONS OF RESULTS, BEGINNING WITH TAXONOMY, MOLECULAR AND MORPHOMETRICS

2) SEE IN THE LEGEND OF TABLE 3. call FIGURE 2 is the correct?? 

3) check the list of references. figures 6, 7, 8, 9 and 10 are not in correct sequence

Author Response

Dear Reviewer,

Thank you very much for your efforts in reviewing the MS.

  • REORDER THE SECTIONS OF RESULTS, BEGINNING WITH TAXONOMY, MOLECULAR AND MORPHOMETRICS. I understand the remark and consider it. Indeed, the suggestion to reorder sections is understandable and corresponds to commonly accepted order. However, I would ask to agree and retain the present order because it reflects the logical set of steps in our research: first, a suspicion that our sampling is heterogeneous; second, verification of the suspicion by morphometric statistics; third, genetic examination of morphometric groups of specimens; taxonomic identification of obtained groups and description of a new species. Taxonomic section contains conclusive results – hence, we would retain it at the end.
  • SEE IN THE LEGEND OF TABLE 3. call FIGURE 2 is the correct??  Legend of table 3 is correct. Legend of figure 2 is correct.
  • check the list of references. figures 6, 7, 8, 9 and 10 are not in correct sequence. According to Rules for Authors, numeration of the references should be continuous including those in tables inserted in the text. Figures 6, 7, 8, 9 and 10 are placed in the Table 1.

Reviewer 4 Report

Dear Authors,

The study “Marine nematodes Acanthopharynx (Nematoda, Chromadoria, Desmodoridae) dwelling in tropical demosponges. Integrative taxonomy with description of a new species” contains a description of two Acanthopharynx marine nematodes, statistical morphometric analysis and molecular genetic analyses. A new species, A. parva sp. n., described are distinct from other known species in the genus. This is a very good paper providing an integrative information on the Acanthopharynx nematodes including a very good description together with fine drawings and high quality pictures. But there are some minor errors in the manuscript. I have added a few comments in the manuscript. For the errors, see the comments on the adjusted manuscript. Please, see the attached file.

Author Response

Dear Reviewer,

We are very thankful for reviewing our MS.

Improved words and sentences are marked with green.

Reviewer 5 Report

I find the paper very clear and prperly presented, including figures (very good). I reckon the present manuscript worth to be published in Diversity, as it is.

If I may, just a refinement of the English language could improve the quality of the manuscript. 

Author Response

Dear Reviewer,

thank you for evaluation of our manuscript. Improved version is attached.

Regards, Alexei Tchesunov
